# EVALUATION METRICS FOR GRAPH GENERATIVE MODELS: PROBLEMS, PITFALLS, AND PRACTICAL SOLUTIONS

**Leslie O'Bray**[1,2,†]**, Max Horn**[1,2,†]**, Bastian Rieck**[1,2,3,4,∗] **and Karsten Borgwardt**[1,2,∗]
[1]Department of Biosystems Science and Engineering, ETH Zürich, Switzerland
[2]SIB Swiss Institute of Bioinformatics, Switzerland
[3]Institute of AI for Health, Helmholtz Munich, Germany
[4]Technical University of Munich, Germany
[†] These authors contributed equally. [∗] These authors jointly supervised this work.

## ABSTRACT

Graph generative models are a highly active branch of machine learning. Given the steady development of new models of ever-increasing complexity, it is necessary to provide a principled way to *evaluate* and *compare* them. In this paper, we enumerate the desirable criteria for such a comparison metric and provide an overview of the status quo of graph generative model comparison in use today, which predominantly relies on the maximum mean discrepancy (MMD). We perform a systematic evaluation of MMD in the context of graph generative model comparison, highlighting some of the challenges and pitfalls researchers inadvertently may encounter. After conducting a thorough analysis of the behaviour of MMD on synthetically-generated perturbed graphs as well as on recently-proposed graph generative models, we are able to provide a suitable procedure to mitigate these challenges and pitfalls. We aggregate our findings into a list of practical recommendations for researchers to use when evaluating graph generative models.

## 1 INTRODUCTION

Graph generative models have become an active research branch, making it possible to generalise structural patterns inherent to graphs from certain domains—such as chemoinformatics—and actively synthesise *new* graphs (Liao et al., 2019). Next to the development of improved models, their *evaluation* is crucial. This is a well-studied issue in other domains, leading to metrics such as the 'Fréchet Inception Distance' (Heusel et al., 2017) for comparing image-based generative models. Graphs, however, pose their own challenges, foremost among them being that an evaluation based on visualisations, i.e. on *perceived* differences, is often not possible. In addition, virtually all relevant structural properties of graphs exhibit spatial invariances, e.g., connected components and cycles are invariant with respect to rotations—that have to be taken into account by a comparison metric.

While the community has largely gravitated towards a single comparison metric, the maximum mean discrepancy (MMD) (Chen et al., 2021; Dai et al., 2020; Goyal et al., 2020; Liao et al., 2019; Mi et al., 2021; Niu et al., 2020; Podda & Bacciu, 2021; You et al., 2018; Zhang et al., 2021), neither its expressive power nor its other properties have been systematically investigated in the context of graph generative model comparison. The goal of this paper is to provide such an investigation, starting from first principles by describing the desired properties of such a comparison metric, providing an overview of what is done in practice today, and systemically assessing MMD's behaviour using recent graph generative models. We highlight some of the caveats and shortcomings of the existing status quo, and provide researchers with practical recommendations to address these issues. We note here that our investigations focus on assessing the structural similarity of graphs, which is a necessary first step before one can jointly assess graphs based on structural and attribute similarity. This paper purposefully refrains from developing its *own* graph generative model to avoid any bias in the comparison.

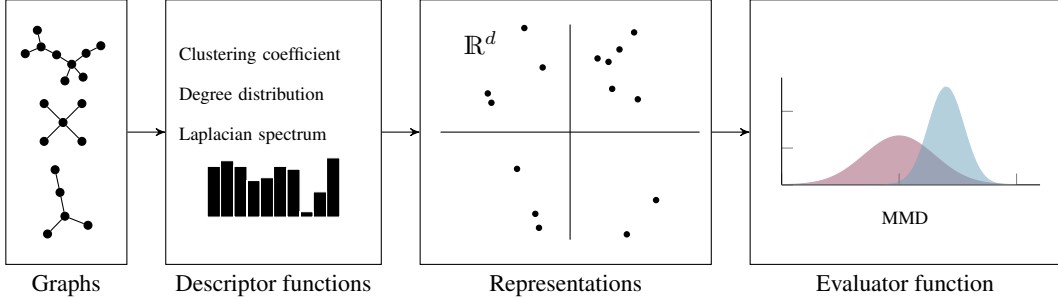

Figure 1: An overview of the workflow used to evaluate graph generative models, as is used, e.g., in Liao et al. (2019); Niu et al. (2020); You et al. (2018): given a distribution of graphs, a set of descriptor functions is employed to map each graph to a high-dimensional representation in $\mathbb{R}^d$. These representations are then compared (with a reference distribution or with each other) using an evaluator function called the maximum mean discrepancy (MMD). In principle, MMD does not require the vectorial representation, but we find that this is the predominant use in the context of graph generative model evaluation.

## 2 COMPARING GRAPH DISTRIBUTIONS

In the following, we will deal with undirected graphs. We denote such a graph as $G = (V, E)$ with vertices $V$ and edges $E$. We treat graph generative models as black-box models, each of which results in a set[1] of graphs $\mathcal{G}$. The original empirical distribution of graphs is denoted as $\mathcal{G}^*$. Given models $\{\mathcal{M}_1, \mathcal{M}_2, \dots\}$, with generated sets of graphs $\{\mathcal{G}_1, \mathcal{G}_2, \dots\}$, the goal of generative model evaluation is to assess which model is a better fit, i.e. which distribution is closer to $\mathcal{G}^*$. This requires the use of a (pseudo-)metric $d(\cdot, \cdot)$ to assess the dissimilarity between $\mathcal{G}^*$ and generated graphs. We argue that the *desiderata* of any such comparison metric are as follows:

1. **Expressivity**: if $\mathcal{G}$ and $\mathcal{G}'$ do not arise from the same distribution, a suitable metric should be able to detect this. Specifically, $d(\mathcal{G}, \mathcal{G}')$ should be monotonically increasing as $\mathcal{G}$ and $\mathcal{G}'$ become increasingly dissimilar.

2. **Robustness**: if a distribution $\mathcal{G}$ is subject to a perturbation, a suitable metric should be robust to small perturbations. Changes in the metric should be ideally upper-bounded by a function of the amplitude of the perturbation. Robust metrics are preferable because of the inherent stochasticity of training generative models.

3. **Efficiency**: model comparison metrics should be reasonably fast to calculate; even though model evaluation is a *post hoc* analysis, a metric should scale well with an increasing number of graphs and an increasing size of said graphs.

While there are many ways to compare two distributions, ranging from statistical divergence measures to proper metrics in the mathematical sense, the comparison of distributions of graphs is exacerbated by the fact that individual graphs typically differ in their cardinalities and are only described up to permutation. With distances such as the graph edit distance being NP-hard in general (Zeng et al., 2009), which precludes using them as an *efficient* metric, a potential alternative is provided by using descriptor functions. A descriptor function $f$ maps a graph $G$ to an auxiliary representation in some space $\mathcal{Z}$. The problem of comparing a generated distribution $\mathcal{G} = \{G_1, \dots, G_n\}$ to the original distribution $\mathcal{G}^* = \{G_1^*, \dots, G_m^*\}$ thus boils down to comparing the *images* $f(\mathcal{G}) := \{f(G_1), \dots, f(G_n)\} \subseteq \mathcal{Z}$ and $f(\mathcal{G}^*) \subseteq \mathcal{Z}$ by any preferred statistical distance in $\mathcal{Z}$ (see Figure 1).

## 3 CURRENT STATE OF GRAPH GENERATIVE MODEL EVALUATION: MMD

Of particular interest in previous literature is the case of $\mathcal{Z} = \mathbb{R}^d$ and using the *maximum mean discrepancy* $d_{\mathrm{MMD}}(\cdot, \cdot)$ as a metric. MMD is one of the most versatile and expressive options available for comparing distributions of structured objects such as graphs (Borgwardt et al., 2006;

---

[1]Formally, the output can also be a multiset, because graphs are allowed to be duplicated.

Table 1: The kernels & parameters chosen by three graph generative models for the MMD calculation.

| Model | | Kernel | Parameter choice $\sigma$ and $n_{\mathrm{bin}}$ | | |
| --- | --- | --- | --- | --- | --- |
| | | | Degree | Clustering | Laplacian |
| Model A | EMD | $\exp\left(W(x,y)/2\sigma^2\right)$ | $\sigma = 1, n_{\mathrm{bin}} = \text{maxdegree}$ | $\sigma = 0.1, n_{\mathrm{bin}} = 100$ | N/A |
| Model B | TV | $\exp\left(-\frac{d_{\mathrm{TV}}(x,y)^2}{2\sigma^2}\right)$ | $\sigma = 1, n_{\mathrm{bin}} = \text{maxdegree}$ | $\sigma = 0.1, n_{\mathrm{bin}} = 100$ | $\sigma = 1, n_{\mathrm{bin}} = 200$ |
| Model C | RBF | $\exp(-\|x-y\|^2/2\sigma^2)$ | $\sigma = 1, n_{\mathrm{bin}} = \text{maxdegree}$ | $\sigma = 0.1, n_{\mathrm{bin}} = 100$ | $\sigma = 1, n_{\mathrm{bin}} = 200$ |

Gretton et al., 2007), providing also a principled way to perform two-sample tests (Bounliphone et al., 2016; Gretton et al., 2012a; Lloyd & Ghahramani, 2015). It enables the comparison of two statistical distributions by means of *kernels*, i.e. similarity measures for structured objects. Letting $\mathcal{X}$ refer to a non-empty set, a function $\mathrm{k}\colon \mathcal{X} \times \mathcal{X} \to \mathbb{R}$ is a kernel if $\mathrm{k}(x_i, x_j) = \mathrm{k}(x_j, x_i)$ for $x_i, x_j \in \mathcal{X}$ and $\sum_{i,j} c_i c_j \, \mathrm{k}(x_i, x_j) \geq 0$ for $x_i, x_j \in \mathcal{X}$ and $c_i, c_j \in \mathbb{R}$. MMD uses such a kernel function to assess the distance between two distributions. Given $n$ samples $X = \{x_1, \dots, x_n\} \subseteq \mathcal{X}$ and $m$ samples $Y = \{y_1, \dots, y_m\} \subseteq \mathcal{X}$, the biased empirical estimate of the MMD between $X$ and $Y$ is obtained as

$$\mathrm{MMD}^2(X, Y) := \frac{1}{n^2} \sum_{i,j=1}^{n} \mathrm{k}(x_i, x_j) + \frac{1}{m^2} \sum_{i,j=1}^{m} \mathrm{k}(y_i, y_j) - \frac{2}{nm} \sum_{i=1}^{n} \sum_{j=1}^{m} \mathrm{k}(x_i, y_j). \quad (1)$$

Since MMD is known to be a metric on the space of probability distributions under certain conditions, Eq. 1 is often treated as a metric as well (Liao et al., 2019; Niu et al., 2020; You et al., 2018).[2] We use the unbiased empirical estimate of MMD (Gretton et al., 2012a, Lemma 6) in our experiments, which removes the self-comparison terms in Equation 1. MMD has been adopted by the community and the current workflow includes two steps: (i) choosing a descriptor function $f$ as described above, and (ii) choosing a kernel on $\mathbb{R}^d$ such as an RBF kernel. One then evaluates $\mathrm{d}_{\mathrm{MMD}}(\mathcal{G}, \mathcal{G}^*) := \mathrm{MMD}(f(\mathcal{G}), f(\mathcal{G}^*))$ for a sample of graphs $\mathcal{G}$. Given multiple distributions $\{\mathcal{G}_1, \mathcal{G}_2, \dots\}$, the values $\mathrm{d}_{\mathrm{MMD}}(\mathcal{G}_i, \mathcal{G}^*)$ can be used to *rank* models: smaller values are assumed to indicate a larger agreement with the original distribution $\mathcal{G}^*$. We will now describe this procedure in more detail and highlight some of its pitfalls.

### 3.1 KERNELS & DESCRIPTOR FUNCTIONS

Before calculating the MMD distance between two samples of graphs, we need to define both the kernel function $\mathrm{k}$ and the descriptor function $f$ that will convert a graph G to a representation in $\mathcal{Z} = \mathbb{R}^d$, for use in the MMD calculation in Eq. 1. We observed a variety of kernel choices in the existing literature. In fact, in three of the most popular graph generative models in use today, which we explore in detail in this paper, a different kernel was chosen for each one. These include a kernel using the first Wasserstein distance (EMD), total variation distance (TV), and the radial basis function kernel (RBF), and are listed in Table 1. In the current use of MMD, a descriptor function $f$ is used to create a vectorial representation of a graph for use in the kernel computation. We find that several descriptor functions are commonly employed, either based on summary statistics of a graph, such as degree distribution histogram and clustering coefficient histogram, or based on spectral properties of the graph, such as the Laplacian spectrum histogram. While several papers also consider the orbit as a descriptor function, we do not consider it in depth here due to its computational complexity, which violates the "efficiency" property from our desiderata. We will now provide brief explanations of these prominent descriptor functions.

**Degree distribution histogram.** Given a graph $\mathrm{G} = (V, E)$, we obtain a histogram by evaluating $\deg(v)$ for $v \in V$, where position $i$ of the resulting histogram is the number of vertices with degree $i$. Assuming a maximum degree $d$ and extending the histogram with zeros whenever necessary, we obtain a mapping $f\colon \mathcal{G} \to \mathbb{R}^d$. This representation has the advantage of being easy to calculate and easy to compare; by normalising it (so that it sums to 1), we obtain a size-invariant descriptor.

---

[2]We will follow this convention in this paper and refer to Eq. 1 as a distance.

**Clustering coefficient.** The (local) clustering coefficient of a vertex $v$ is defined as the fraction of edges within its neighbourhood divided by the number of all possible edges between neighbours, i.e.

$$C(v) := \frac{2|\{(v_i, v_j) \in E \mid v_i \in \mathcal{N}(v) \vee v_j \in \mathcal{N}(v)\}|}{\deg(v)(\deg(v) - 1)}. \tag{2}$$

The value of $C(v) \in [0, 1]$ measures to what extent a vertex $v$ forms a clique (Watts & Strogatz, 1998). The collection of all clustering coefficients of a graph can be binned and converted into a histogram in order to obtain a graph-level descriptor. This function is also easy to calculate but is inherently local; a graph consisting of disconnected cliques or a fully-connected graph cannot be distinguished, for example.

**Laplacian spectrum histogram.** Spectral methods involve assigning a matrix to a graph G, whose spectrum, i.e. its eigenvalues and eigenvectors, is subsequently used as a characterisation of G. Let $\mathbf{A}$ refer to the *adjacency matrix* of G, with $\mathbf{A}_{ij} = 1$ if and only if vertices $v_i$ and $v_j$ are connected by an edge in G (since G is undirected, $\mathbf{A}$ is symmetric). The *normalised graph Laplacian* is defined as $\mathcal{L} := \mathbf{I} - \mathbf{D}^{-\frac{1}{2}} \mathbf{A} \mathbf{D}^{-\frac{1}{2}}$, where $\mathbf{I}$ denotes the identity matrix and $\mathbf{D}$ refers to the *degree matrix*, i.e. $\mathbf{D}_{ii} = \deg(v_i)$ for a vertex $v_i$ and $\mathbf{D}_{ij} = 0$ for $i \neq j$. The matrix $\mathcal{L}$ is real-valued and symmetric, so it is diagonalisable with a full set of eigenvalues and eigenvectors. Letting $\lambda_1 \leq \lambda_2 \leq \dots$ refer to the eigenvalues of $\mathcal{L}$, we have $0 \leq \lambda_i \leq 2$ (Chung, 1997, Chapter 1, Lemma 1.7). This boundedness lends itself naturally to a histogram representation (regardless of the size of G), making it possible to bin the eigenvalues and use the resulting histogram as a simplified descriptor of a graph. The expressivity of such a representation is not clear a priori; the question of whether graphs are fully determined by their spectrum is still open (van Dam & Haemers, 2003) and has only been partially answered *in the negative* for certain classes of graphs (Schwenk, 1973).

## 4 ISSUES WITH THE CURRENT PRACTICE

Common practice in graph generative model papers is to use MMD by fixing a kernel and parameter values for the descriptor functions and kernel, and then assessing the newly-proposed model as well as its competitor models using said kernel and parameters. Authors chose different values of $\sigma$ for the different descriptor functions, but to the best of our knowledge, all parameters are set to a fixed value a priori *without* any selection procedure. If MMD were able to give results and rank different models in a stable way across different kernel and parameter choices, this would be inconsequential. However, as our experiments will demonstrate, the results of MMD are *highly sensitive* to such choices and can therefore lead to an arbitrary ranking of models.

Subsequently, we use three current real-world models, GraphRNN (You et al., 2018), GRAN (Liao et al., 2019), and Graph Score Matching (Niu et al., 2020). We ran the models using the author-provided implementations to generate new graphs on the Community, Barabási-Albert, Erdös-Rényi, and Watts-Strogatz graph datasets, and then calculated the MMD distance between the generated graphs and the test graphs, using the different (i) kernels that they used (EMD, TV, RBF), (ii) descriptor functions (degree histogram, clustering coefficient histogram, and Laplacian spectrum), and (iii) parameter ranges ($\sigma, \lambda \in \{10^{-5}, \dots, 10^5\}$). For simplicity, we will refer to the parameter in all kernels as $\sigma$. We purposefully *refrained from using model names*, preferring instead 'A, B, C' in order to focus on the issues imposed by such an evaluation, rather than providing a commentary on the performance of a specific model.

In the following, we will delve deeper into issues originating from the individual components of the graph generative model comparison in use today. Due to space constraints, examples are provided on individual datasets; full results across datasets are available in Appendix A.6.

### 4.1 NUANCES OF USING MMD FOR GRAPH GENERATIVE MODEL EVALUATION

While MMD may seem like a reasonable first choice as a metric for comparing distributions of graphs, it is worth mentioning two peculiarities of such a choice and how authors are currently applying it in this context. First, MMD was originally developed as an approach to perform two-sample testing on structured objects such as graphs. As such, its suitability was investigated in that context, and *not* in that of evaluating graph generative models. This warrants an investigation of the implications of "porting" such a method from one context to another.

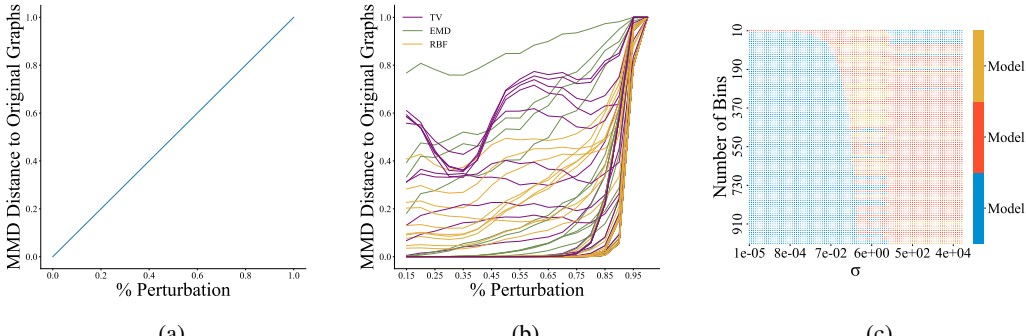

(a)             (b)             (c)

Figure 2: Figure 2a shows the ideal behaviour of a graph generative model evaluator: as two distributions of graphs become increasingly dissimilar, e.g. via perturbations, the metric should grow proportionally. Figure 2b shows the behaviour of the current choices in reality; each line represents the normalized MMD for a given kernel and parameter combination. A cautious choice of kernel and parameters is needed in order to obtain a metric with behaviour similar to Figure 2a. Each square in Figure 2c shows which model performs best (out of A, B, and C) over a grid of hyperparameter combinations of $\sigma$ and number of bins in the histogram. Any model can rank first with an appropriate hyperparameter selection, showcasing the sensitivity of MMD to the hyperparameter choice.

The second peculiarity worth mentioning is that MMD was groundbreaking for its ability to compare distributions of structured objects by means of a kernel, thus bypassing the need to employ an intermediate vector representation. Yet, in its current application, the graphs are first being vectorised, and then MMD is used to compare the vectors. MMD is not technically required in this case: any statistical method for comparing vector distributions could be employed, such as Optimal Transport (OT). While the use of different evaluators besides MMD for assessing graph generative models is an interesting area for future research, we focus specifically on MMD, since this is what is in use today, and now highlight two practical issues that can arise from using MMD in this context.

**MMD's ability to capture differences between distributions is kernel- and parameter-dependent.** As two distributions become sufficiently dissimilar, we find that their distance should monotonically increase as a function of their dissimilarity. While one can construct specific scenarios in which two distributions become farther apart but the distance does not monotonically increase (e.g., removing edges from a triangle-free graph, and using the clustering coefficient as the descriptor function), in such cases, the specific choice of descriptor function $f$ is crucial to ensure that $f$ can capture differences in distribution. However, it is not guaranteed that MMD will monotonically increase as two distributions become increasingly dissimilar. Figure 2b depicts this behaviour when focusing on a single descriptor function, the clustering coefficient, with each line representing a unique kernel and parameter choice (see Appendix A.6 for the full results for other datasets and descriptor functions). We subject an original set of graphs to perturbations of increasing magnitude and then measured the MMD distance to the original distribution. Despite both distributions becoming progressively dissimilar by experimental design, a large number of kernel/parameter configurations *fail* to capture this, showing that MMD is highly sensitive to this choice. In many instances, the distance remains nearly constant, despite the increased level of perturbation, until the magnitude of the perturbation reaches an extraordinarily high level. In some cases, we observe that the distance even *decreases* as the degree of perturbation *increases*, suggesting that the original data set and its perturbed variant are more similar. We also find that the MMD values as a function of the degree of perturbation are highly sensitive to the kernel and parameter selection, as evidenced by the wide range of different curve shapes observed in Figure 2b.

**MMD has no inherent scale.** Another challenge with MMD is that since current practice works with the raw MMD distance, as opposed to p-values, as originally proposed in Gretton et al. (2012a), there is no inherent scale of the MMD values. It is therefore difficult to assess whether the smaller MMD distance of one model is *substantially* improved when compared to the MMD distance of another model. For instance, suppose that the MMD distance of one model is $5.2 \times 10^{-8}$. Is this a

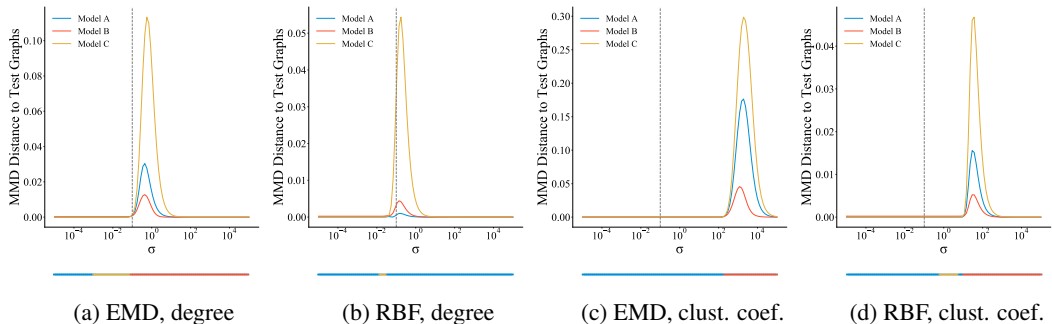

Figure 3: This shows the MMD distance to the test set of graphs for three recent graph generative models (whose names we intentionally omitted) on the Community Graphs dataset for different descriptor functions and kernels. MMD requires the choice of a kernel and kernel parameters. Each subfigure shows MMD (lower is better) along a range of values of $\sigma$ (reported on a $\log$ scale), with the bar underneath indicating which model ranks first for the given value of $\sigma$. The grey line indicates the $\sigma$ chosen by the authors. Subfigures 3a and 3b show how simply switching from the EMD to the RBF kernel (holding $\sigma$ constant) can change which model performs best; Subfigures 3c and 3d show how the choice of $\sigma$ by the authors misses the area of maximum discrimination of MMD.

meaningfully better model than one whose MMD distance is $4.6 \times 10^{-7}$? This is further compounded by the choice of kernel and parameter, which also affects the scale of MMD. Since common heuristics can lead to a suboptimal choice of $\sigma$ for MMD (Sutherland et al., 2017), authors may obtain an arbitrarily low value of MMD, as seen in Figure 3c and 3d. As MMD results are typically reported just in a table, the lack of a scale hinders the reader's ability to effectively assess a model's performance.

## 4.2 CONSEQUENCES OF THE CHOICE OF KERNEL

MMD requires the choice of a kernel function, yet there is no agreed-upon method to select one. We find that each of the three models considered in this paper choose a different kernel for its evaluation, as evidenced in Table 1. There are several potential issues with the current practice, which we will subsequently discuss, namely (i) the selected kernel might be computationally inefficient to compute, as is the case for the previously-used Earth Mover's Distance (EMD) kernel function, (ii) the use of positive definite kernel functions is a limitation, with previous work employing functions that are *not* valid kernels, and (iii) finally, the kernel choice may result in arbitrary rankings, and there is currently little attention paid to this choice.

**Computational cost of kernel computation.** Issue (i) might prevent an evaluation metric to be used in practice, thus stymieing graph generative model development. While the choice of a kernel using the first Wasserstein distance is a valid choice, it is extremely slow to compute, as noted by Liao et al. (2019). From this perspective, it violates the third quality of our desiderata: efficiency.

**Kernels need to be p.s.d.** To reduce the aforementioned computational costs, previous work (Liao et al., 2019) used a kernel based on the total variation distance between histograms, which has since been used in subsequent publications as one of the ways to evaluate different models. As we show in Appendix A.1, this approach leads to an *indefinite kernel* (i.e. the kernel is neither positive definite nor negative definite), whose behaviour in the context of MMD is not well-defined. MMD necessitates the use of p.s.d. kernels, and care must be taken when it comes to interpreting the respective results.

**Arbitrary ranking based on kernel choice.** Issue (iii) relates to the fact that changing the kernel can lead to different results of model evaluation, which is problematic since each paper we considered used a different kernel. For instance, with the degree distribution as the graph descriptor function, simply changing the choice of kernel from EMD to RBF (while holding the parameters constant) leads to a different ranking of the models, which can be seen in Figure 3a–Figure 3b, where the best performing model changes merely by changing the kernel choice (!). This type of behaviour is highly undesired and problematic, as it implies that model performance (in terms of the evaluation) can be improved by choosing a different kernel.

### 4.3 EFFECT OF THE CHOICE OF HYPERPARAMETERS

While the choice of which kernel to use in the MMD calculation is itself an *a priori* design choice without clear justification, it is further exacerbated by the fact that many kernels require picking parameters, without any clear process by which to choose them. To the best of our knowledge, this selection of parameters is glossed over in publications at present.[3] For instance, Table 1 shows how authors are setting the value of $\sigma$ differently for different descriptor functions, yet there is no discussion nor established best practice of *how* such parameters were or should be set. Hyperparameter selection is known to be a crucial issue in machine learning that mandates clear selection algorithms in order to avoid biasing the results. If the choice of parameters—similar to the choice of kernel—had no bearing on the outcome, this would not be problematic, but our empirical experiments prove that drastic differences in model evaluation performance can occur.

**Arbitrary ranking based on parameter choice.** The small colour bars underneath each plot of Figure 3 show the model that achieves the lowest MMD for a given value of $\sigma$. Changes in colour highlight a sensitivity to specific parameter values. Even though the plots showing MMD seem to have a general trend of which model is best in the peak of the curves, in the other regions of $\sigma$, the ranking switches, with the effect that a different model would be declared "best." This is particularly the case in Subfigures 3a, 3c, and 3d, where Model B appears to be the best, yet for much of $\sigma$, a different model ranks first. This sensitivity is further exacerbated by the fact that some descriptor functions require a parameter choice as well, such as the bin size, $n_{\mathrm{bin}}$, for a histogram. Figure 2c shows how the best-ranking model on the Barabási-Albert Graphs is entirely dependent upon the choice of parameters $(n_{\mathrm{bin}}, \sigma)$. The colour in each grid cell corresponds to the best-ranking model for a given $\sigma$ and $n_{\mathrm{bin}}$; we find that this is wildly unstable across both $n_{\mathrm{bin}}$ and $\sigma$. The consequence of this is alarming: *any* model could rank first if the right parameters are chosen.

**Choice of $\sigma$ by authors does *not* align with maximum discrimination in MMD.** Figures 3c and 3d show MMD for different values of $\sigma$ with the clustering coefficient descriptor function in the Community Graphs dataset for Models A, B and C. The value of $\sigma$ as selected by the authors (without justification or discussion) is indicated by the grey line; however, this choice corresponds to a regions of low activity in the MMD curve, suggesting a poor parameter choice. While Model B seems to be the clear winner, the choice of $\sigma$ by the authors resulted in Model A having the best performance. Furthermore, it is not clear that choosing the same $\sigma$ across different kernels, as is currently done, makes sense; whereas $\sigma = 10^3$ would be sensible for EMD in Figure 3c, for the Gaussian kernel in Figure 3d, such a choice too far beyond the discriminative peak.

## 5 HOW TO USE MMD FOR GRAPH GENERATIVE MODEL EVALUATION

Having understood the potential pitfalls of using MMD, we now turn to suggestions on how to better leverage MMD for graph generative model evaluation.

**Provide a sense of scale.** As mentioned in Section 4.1, MMD does not have an inherent scale, making it difficult to assess what is 'good.' To endow their results with some meaning, practitioners should calculate MMD between the test and training graphs, and then include this in the results table/figures alongside the other MMD results. This will provide a meaningful bound on what two 'indistinguishable' sets of graphs look like in a given dataset (see Figures 10–13 in Appendix A.6).

**Choose valid and efficient kernel candidates.** We recommend to avoid the EMD-based kernel due to the computational burden (see Appendix A.8), and the total variation kernel for its non-p.s.d nature. Instead, we suggest using either an RBF kernel, since it is a universal kernel, or a Laplacian kernel, or a linear kernel, i.e. the canonical inner product on $\mathbb{R}^d$, since it is parameter-free. As all of these kernels are p.s.d.,[4] and are fast to compute, they satisfy the efficiency desiderata criteria, and thus only require analysis of their expressivity and robustness.

---

[3]We remark that literature outside the graph generative modelling domain describes such choices, for example in the context of two-sample tests (Gretton et al., 2012b) or general model criticism (Sutherland et al., 2017). We will subsequently discuss to what extent the suggested parameter selection strategies may be transferred.

[4]In the case of the Laplacian kernel, the TV distance in lieu of the Euclidean distance leads to a valid kernel.

**Utilize meaningful descriptor functions.** Different descriptor functions measure different aspects of the graph and are often domain-specific. As a general recommendation, we propose using previously-described (Liao et al., 2019; You et al., 2018) graph-level descriptor functions, namely (i) the degree distribution, (ii) the clustering coefficient, and (iii) the Laplacian spectrum histograms, and recommend that the practitioner make domain-specific adjustments based on what is appropriate.

## 5.1 SELECTING AN APPROPRIATE KERNEL AND HYPERPARAMETERS

The kernel choice and descriptor functions require hyperparameter selection. We recommend assessing the performance of MMD in a controlled setting to elucidate some of the properties specific to the dataset and the descriptor functions of interest to a given application. In doing so, it becomes possible to choose a kernel and parameter combination that will yield informative results. Notice that in contrast to images, where visualisation provides a meaningful evaluation of whether two images are similar or not, graphs cannot be assessed in this manner. It is thus necessary to have a principled approach where the degree of difference between two distributions can be controlled. We subject a set of graphs to perturbations (edge insertions, removals, etc.) of increasing magnitude, thus enabling us to assess the expected degree of difference to the original graphs.

We ideally want an evaluation metric to effectively reflect the degree of perturbations. Hence, with an increasing degree of perturbation of graphs, the distance to the original distribution $\mathcal{G}^*$ of unperturbed graphs should increase. We can therefore assess both the expressivity of the evaluation metric, i.e., its ability to distinguish two distributions when they are different, and its robustness (or stability) based on how rapidly such a metric changes when subject to small perturbations. Succinctly, we would like to see a clear correlation of the metric with the degree of perturbation, thus indicating both robustness and expressivity. Perturbation experiments are particularly appealing because they do not require access to other models but rather only an initial distribution $\mathcal{G}^*$. This procedure therefore does not leak any information from models and is unbiased. Moreover, researchers have more control over the ground truth in this scenario, as they can adjust the desired degree of dissimilarity. While there are many perturbations that we will consider (adding edges, removing edges, rewiring edges, and adding connected nodes), we focus primarily on progressively adding or removing edges. This is the graph analogue to adding "salt-and-pepper" noise to an image, and in its most extreme form (100% perturbation) corresponds to a fully-connected graph and fully-disconnected graph, respectively.

**Creating dissimilarity via perturbations.** For each perturbation type, i.e., (i) random edge insertions, (ii) random edge deletions, (iii) random rewiring operations ('swapping' edges), (iv) random node additions, we progressively perturb the set of graphs, using the relevant perturbation parameters, in order to obtain multiple sets of graphs that are increasingly dissimilar from the original set of graphs. Each perturbation is parametrised by at least one parameter. When removing edges, for instance, the parameter is the probability of removing an edge in the graph. Thus for a graph with 100 edges and $p_{\text{remove}} = 0.1$ we would expect on average 90 edges to remain in the graph. Similar parametrisations apply for the other perturbations, i.e. the probability of adding an edge for edge insertions, the probability of rewiring an edge for edge rewiring, and the number of nodes to add to a graph, as well as the probability of an edge between the new node and the other nodes in the graph for adding connected nodes. We provide a formal description of each process in Appendix A.2 and A.3.

**Correlation analysis to choose a kernel and hyperparameters.** For each of the aforementioned perturbation types, we compared the graph distribution of the perturbed graphs with the original graphs using the MMD. We repeated this for different scenarios, comprising different kernels, different descriptor functions, and where applicable, parameters. For a speed up trick to efficiently calculate MMD over a range of $\sigma$, please see Appendix A.5. Since these experiments resulted in hundreds of configurations, due to the choice of kernel, descriptor function, and parameter choices, we relegated most of the visualisations to the Appendix (see Section A.6). To compare the different configurations effectively, we needed a way to condense the multitude of results into a more interpretable and comparable visualisation. We therefore calculated Pearson's correlation coefficient between the degree of perturbation and the resulting MMD distance, obtaining two heatmaps. The first one shows the *best parameter choice*, the second one shows the *worst parameter choice*, both measured in terms of Pearson's correlation coefficient (Figure 4). In the absence of an agreed-upon procedure to choose such parameters, the heatmaps effectively depict the extremes of what will happen if one is particularly "lucky" or "unlucky" in the choice of parameters. A robust combination of descriptor function and

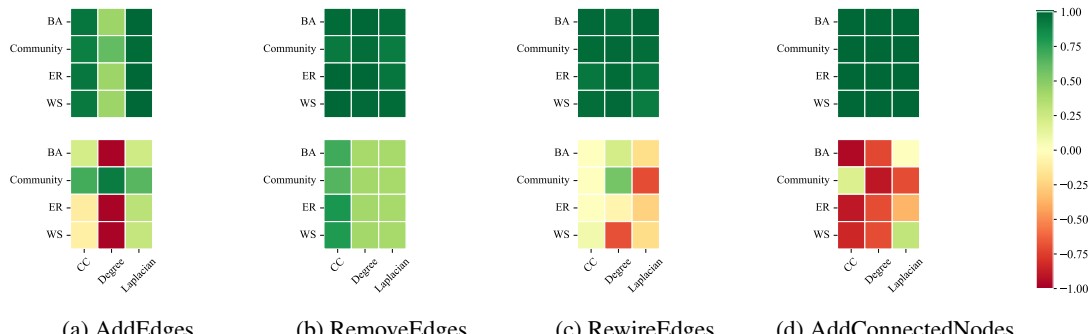

(a) AddEdges  (b) RemoveEdges  (c) RewireEdges  (d) AddConnectedNodes

Figure 4: The correlation of MMD with the degree of perturbation in the graph, assessed for different descriptor functions and datasets (BA: Barabási-Albert, ER: Erdös-Rényi, WS: Watts-Strogatz). For an ideal metric, the distance would increase with the degree of perturbation, resulting in values $\approx 1$. The upper row shows the *best* kernel-parameter combination; the bottom row shows the *worst*. A proper kernel and parameter selection leads to strong correlation to the perturbation, but a bad choice can lead to inverse correlation, highlighting the importance of a good kernel/parameter combination.

comparison function is characterised by both heatmaps exhibiting high correlation with the degree of perturbation. In the bottom row, the MMD distance in many cases no longer shows *any correlation* with the degree of perturbation, and in the case of the clustering coefficient (CC), is even negatively correlated with the degree of perturbation. Such behaviour is undesired, showcasing a potential pitfall for authors if they inadvertently fail to pick a "good" parameter or kernel combination. While we chose the Pearson correlation coefficient for its simplicity and interpretability, other measures of dependence could be used instead if they are domain-appropriate (see Appendix A.7). To choose a kernel and parameter combination, we suggest picking the parameters with the highest correlation for the perturbation that is most meaningful in the given domain. Lacking this, the practitioner could choose the combination that has the highest *average* correlation across perturbations.

## 6 CONCLUSION

We provided a thorough analysis of how graph generative models are being currently assessed by means of MMD. While MMD itself is powerful and expressive, its use has certain idiosyncratic issues that need to be avoided in order to obtain fair and reproducible comparisons. We highlighted some of these issues, most critical of which are that the choice of kernel and parameters can result in different rankings of different models, and that MMD may not monotonically increase as two graph distributions become increasingly dissimilar. As a mitigation strategy, we propose running a perturbation experiment as described in this paper to select a kernel and parameter combination that is highly correlated with the degree of perturbation. This way, the choice of parameters does not depend on the candidate models but only on the initial distribution of graphs.

**Future work.** This work gives an overview of the current situation, illuminates some issues with the status quo, and provides practical solutions. We hope that this will serve as a starting point for the community to further develop methods to assess graph generative models, and it is encouraging that some efforts to do so are already underway (Thompson et al., 2022). Future work could investigate the use of efficient graph kernels in combination with MMD, as described in a recent review (Borgwardt et al., 2020). This would reduce the comparison pipeline in that graph kernels can be directly used with MMD, making graph descriptor functions unnecessary. Another approach could be to investigate alternative or new descriptor functions, such as the geodesic distance, or alternative evaluation methods, such as the multivariate Kolmogorov–Smirnov test (Justel et al., 1997), or even develop totally novel evaluation strategies, preferably those that go beyond the currently-employed vectorial representations of graphs.

## Reproducibility Statement

We have provided the code for our experiments in order to make our work fully reproducible. Details can be found in Appendix A.2-A.4, which includes a link to our GitHub repository. During the review process our code is available as a Supplementary Material, in order to preserve anonymity.

### Acknowledgments

This work was supported in part by the Alfried Krupp Prize for Young University Teachers of the Alfried Krupp von Bohlen und Halbach-Stiftung (K.B.).

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

# A  APPENDIX

The following sections provide additional details about the issues with existing methods. We also show additional plots from our ranking and perturbation experiments.

## A.1  KERNELS BASED ON TOTAL VARIATION DISTANCE

Previous work used kernels based on the total variation distance in order to compare evaluation functions via MMD. The choice of this distance, however, requires subtle changes in the selection of kernels for MMD—it turns out that the usual RBF kernel must *not* be used here!

We briefly recapitulate the definition of the total variation distance before explaining its use in the kernel context: given two finite-dimensional real-valued histograms $X := \{x_1, \ldots, x_n\}$ and $Y := \{y_1, \ldots, y_n\}$, their *total variation distance* is defined as

$$d_{\mathrm{TV}}(X, Y) := \frac{1}{2} \sum_{i=1}^{n} |x_i - y_i|. \tag{3}$$

This distance induces a metric space that is not *flat*. In other words, the metric space induced by Eq. 3 has non-zero curvature (a fact that precludes certain kernels from being used together with Eq. 3). We can formalise this by showing that the induced metric space *cannot* have a bound on its curvature.

**Theorem 1.** *The metric space $X_{\mathrm{TV}}$ induced by the total variation distance between two histograms is not in $\mathrm{CAT}(k)$ for $k > 0$, where $\mathrm{CAT}(k)$ refers to the category of metric spaces with curvature bounded from above by $k$ (Gromov, 1987).*

*Proof.* Let $x_1 = (1, 0, \ldots, 0)$ and $x_2 = (0, 1, 0, \ldots, 0)$. There are at least two geodesics—shortest paths—of the same length, one that first decreases the first coordinate and subsequently increases the second one, whereas for the second geodesic this order is switched. More precisely, the first geodesic proceeds from $x_1$ to $x_1 - (\epsilon, 0, \ldots, 0)$ for an infinitesimal $\epsilon > 0$, until $x_0 = (0, \ldots, 0)$ has been reached. Following this, the geodesic continues from $x_0$ to $x_0 + (0, \epsilon, 0, \ldots, 0)$ in infinitesimal steps until $x_2$ has been reached. The order of these two operations can be switched, such that the geodesic goes from $x_1$ to $(1, 1, 0, \ldots, 0)$, from which it finally continues to $x_2$. Both of these geodesics have a length of 1. Since geodesics in a $\mathrm{CAT}(k)$ space for $k > 0$ are *unique* (Bridson & Haefliger, 1999, Proposition 2.11, p. 23), $X_{\mathrm{TV}}$ is not in $\mathrm{CAT}(k)$ for $k > 0$. $\qquad\square$

Since every $\mathrm{CAT}(k)$ space is also a $\mathrm{CAT}(l)$ space for all $l > k$, this theorem has the consequence that $X_{\mathrm{TV}}$ cannot be a $\mathrm{CAT}(0)$ space. Moreover, as every *flat* metric space is in particular a $\mathrm{CAT}(0)$ space, $X_{\mathrm{TV}}$ is not flat. According to Theorem 1 of Feragen et al. (2015), the associated geodesic Gaussian kernel, i.e. the kernel that we obtain by writing

$$k(x, y) := \exp\left(-\frac{d_{\mathrm{TV}}(X, Y)^2}{2\sigma^2}\right), \tag{4}$$

is *not* positive definite and should therefore not be used with MMD. One potential fix for this specific distance involves using the Laplacian kernel, i.e.

$$k(x, y) := \exp(-\lambda \, d_{\mathrm{TV}}(X, Y)). \tag{5}$$

The subtle difference between these kernel functions—only an exponent is being changed—demonstrate that care must be taken when selecting kernels for use with MMD.

## A.2  EXPERIMENTAL SETUP

We can analyse the desiderata outlined above using an experimental setting. In the following, we will assess expressivity, robustness, and efficiency for a set of common perturbations, i.e. (i) random edge insertions, (ii) random edge deletions, (iii) random rewiring operations, i.e. 'swapping' edges, and (iv) random node additions. For each perturbation type, we will investigate how the metric changes for an ever-increasing degree of perturbation, where each perturbation is parametrized by at least one parameter. When removing edges, for instance, this is the probability of removing an edge in the graph. Thus for a graph with 100 edges and $p_{\mathrm{remove}} = 0.1$ we would expect on average 90 edges to

remain in the graph. Similar parametrizations apply for the other perturbations, for anmore detailed description we refer to Appendix A.3. All these operations are inherently small-scale (though not necessarily localised to specific regions within a graph), but turn into large-scale perturbations of a graph depending on the strength of the perturbation performed. We performed these perturbations using our own `Python`-based framework for graph generative model comparison and evaluation. Our framework additionally permits the simple integration of additional descriptor functions and evaluators.

## A.3  DETAILS ON GRAPH PERTURBATIONS

We describe all graph perturbations used in this work on the example of a single graph $G := (V, E)$ where $V$ refers to the vertices of the graph and $E$ to the edges.

**Add Edges**  For each $v_i, v_j \in V$ with $v_i \neq v_j$ a sample from a Bernoulli distribution $x_{ij} \sim \text{Ber}(p_{\text{add}})$ is drawn. Samples for which $x_{ij} = 1$ are added to the list of edges such that $E' = E \cup \{(v_i, v_j) \mid x_{ij} = 1\}$.

**Remove Edges**  For each $e_i \in E$, a sample from a Bernoulli distribution $x_i \sim \text{Ber}(p_{\text{remove}})$ is drawn, and samples with $x_i = 1$ are removed from the edge list, such that $E' = E \cap \{e_i \mid x_i \neq 1\}$.

**Rewire Edges**  For each $e_i \in E$, a sample from a Bernoulli distribution $x_i \sim \text{Ber}(p_{\text{rewire}})$ is drawn, and samples with $x_i = 1$ are rewired. For rewiring a further random variable $y_i \sim \text{Ber}(0.5)$ is drawn which determines which node $e_i[y_i]$ of the edge $e_i$ is kept. The node to which the edge is connected is chosen uniformly from the set of vertices $v_i \in V$, where $v_i \notin e_i$, i.e. avoiding self-loops and reconnecting the original edge. Finally the original edge is removed and the new edge $e'_i = (e_i[y_i], v_i)$ is added to the graph $E' = E \cap \{e_i \mid x_i \neq 1\} \cup \{e'_i \mid x_i = 1\}$.

**Add Connected Node**  We define a set of vertices to be added $V^* = \{v_i \mid |V| < i \leq |V| + n\}$, where $n$ represents the number of nodes to add. For each $v_i \in V$ and $v_j \in V^*$ we draw a sample from a Bernoulli distribution $x_{ij} \sim \text{Ber}(p_{\text{connect\_node}})$ and an edge between $v_i$ and $v_j$ to the graph if $x_{ij} = 1$. Thus $E' = E \cup \{(v_i, v_j) \mid v_i \in V, v_j \in V^*, x_{ij} = 1\}$.

## A.4  IMPLEMENTATION DETAILS

We used the official implementations of GraphRNN, GRAN and Graph Score Matching in our experiments. GraphRNN and GRAN both have an MIT License, and Graph Score Matching is licensed under GNU General Public License v3.0. Our code is available at (`https:/www.github.com/BorgwardtLab/ggme`) under a BSD 3-Clause license.

**Compute resources.**  All the jobs were run on our internal cluster, comprising 64 physical cores (`Intel(R) Xeon(R) CPU E5-2620 v4 @ 2.10GHz`) with 8 GeForce GTX 1080 GPUs. We stress that the main component of this paper, i.e. the evaluation itself, do *not* necessarily require a cluster environment. The cluster was chosen because individual generative models had to be trained in order to obtain generated graphs, which we could subsequently analyse and rank.

## A.5  SPEED UP TRICK

As the combination of kernels and hyperparameters yielded hundreds of combinations, it is worth mentioning a worthwhile speedup trick to reduce the complexity of assessing the Gaussian and Laplacian kernel combinations. Considering they have a shared intermediate value, namely the Euclidean distance, it is possible to return intermediate values in the MMD computation ($K_{XX}$, $K_{YY}$, and $K_{XY}$, prior to exponentiation, potential squaring, and scaling by $\sigma$. Storing these intermediate results allows one to rapidly iterate over a grid of values for $\sigma$ without needed to recalculate MMD, leading to a worthwhile speedup.

## A.6  EXPERIMENTAL RESULTS

We now provide the results presented in the paper across all the datasets.

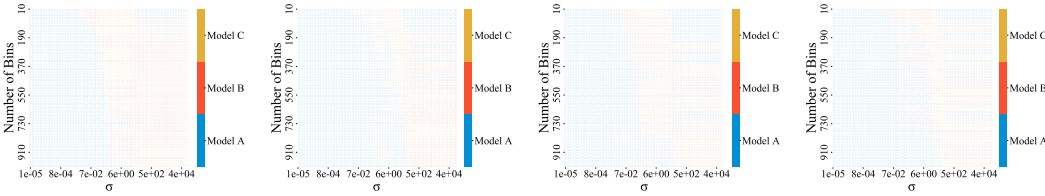

(a) Barabási-Albert Graphs     (b) Community Graphs     (c) Erdös-Rényi Graphs     (d) Watts-Strogatz Graphs

Figure 5: A heatmap of which model (from A, B, and C) ranks first in terms of MMD across different hyperparameter combinations. This uses the clustering coefficient descriptor function and the RBF kernel, where the number of bins is a hyperparameter of the descriptor function, and $\sigma$ is the hyperparameter in the kernel.

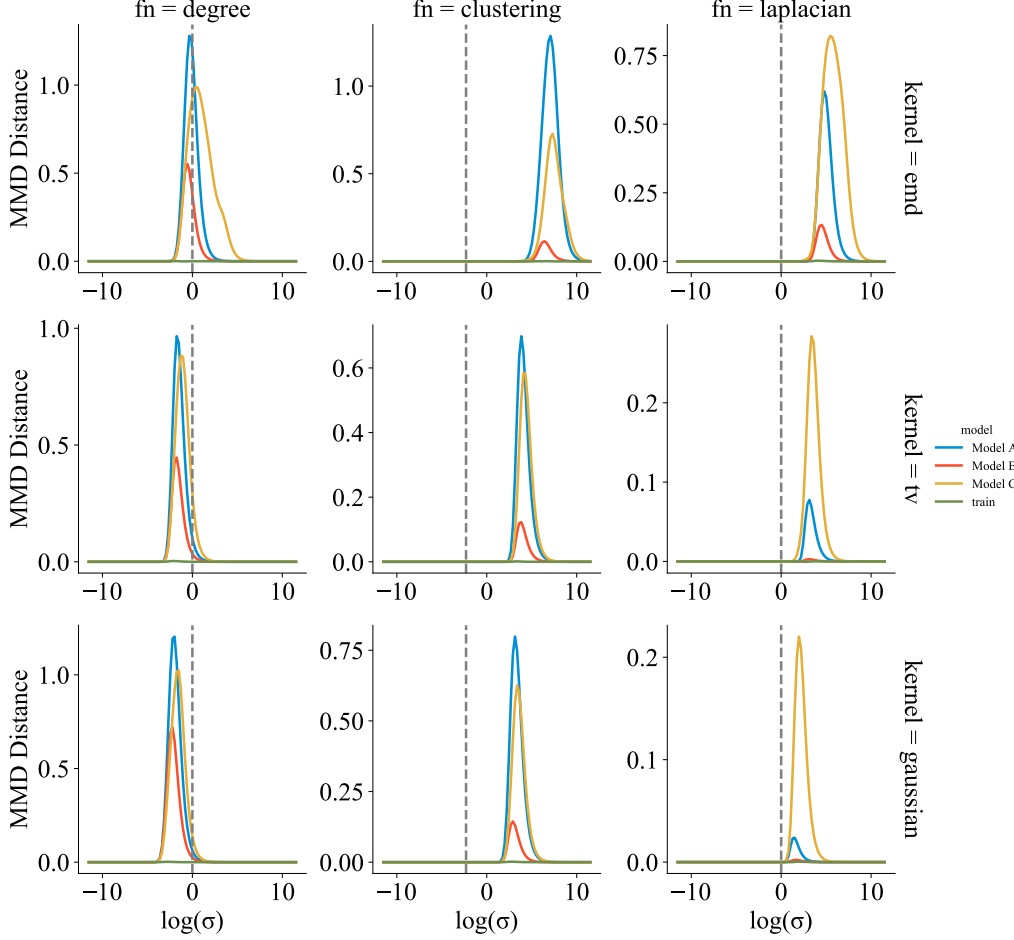

Figure 6: **Barabási-Albert Graphs**. MMD calculated between the test graphs and predictions from three recent graph generative models (A, B, C) over a range of values of $\sigma$ on the Barabási-Albert Graphs dataset. Additionally, the MMD distance between the test graphs and training graphs is provided to give a meaningful sense of scale to the metric. It provides an idea of what value of MMD signifies an indistinguishable difference between the two distributions.

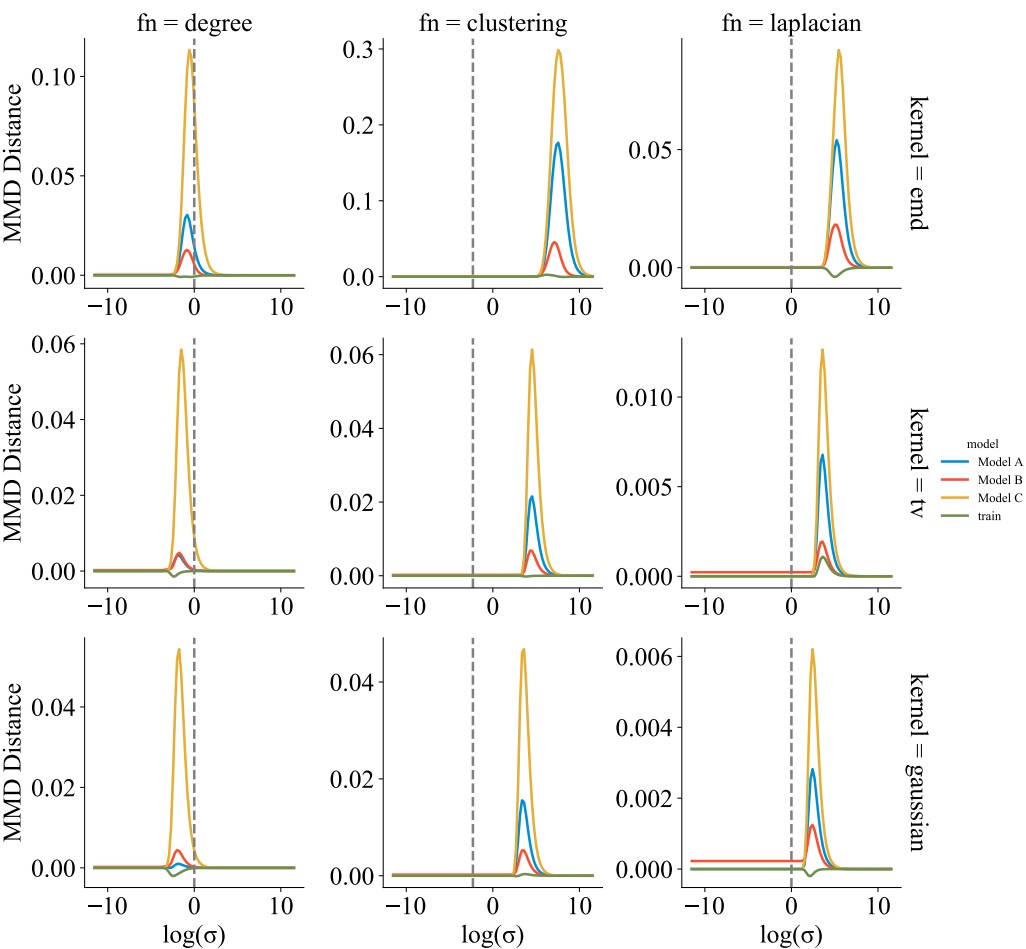

Figure 7: **Community Graphs**. MMD calculated between the test graphs and predictions from three recent graph generative models (A, B, C) over a range of values of $\sigma$ on the Community Graphs dataset. Additionally, the MMD distance between the test graphs and training graphs is provided to give a meaningful sense of scale to the metric. It provides an idea of what value of MMD signifies an indistinguishable difference between the two distributions.

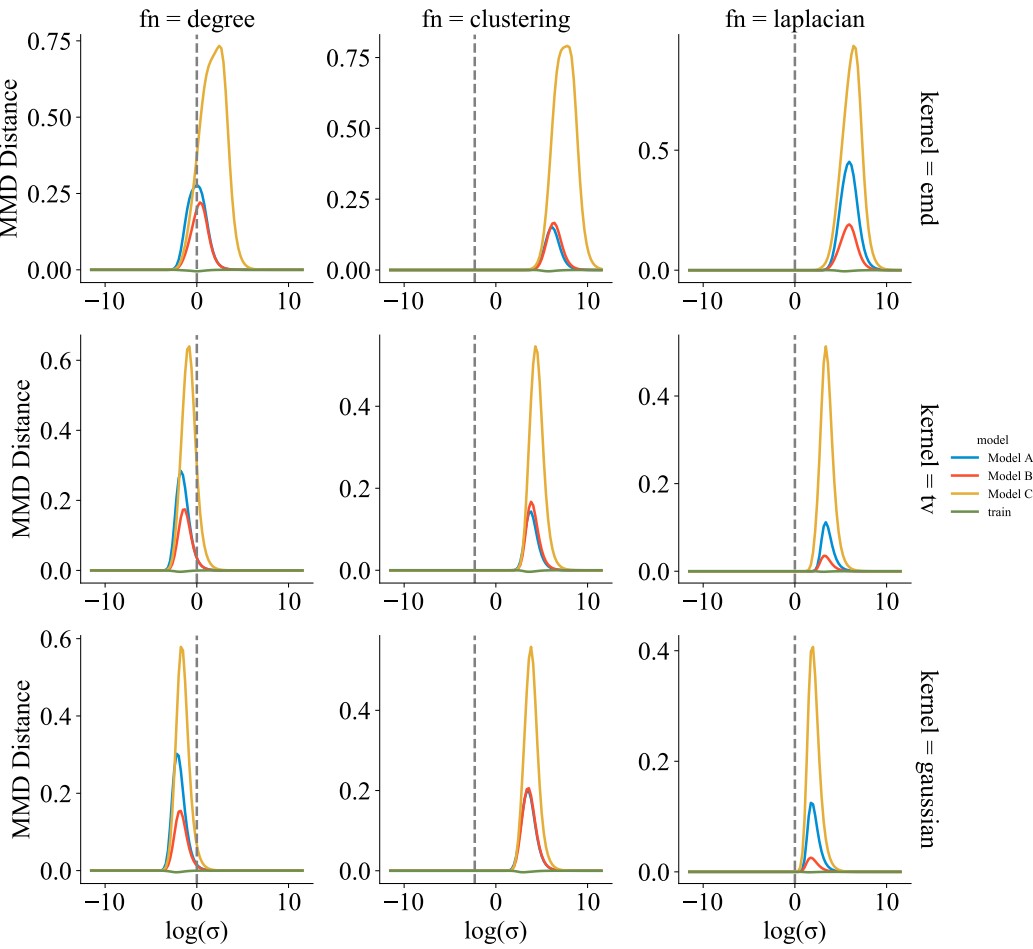

Figure 8: **Erdös-Rényi Graphs**. MMD calculated between the test graphs and predictions from three recent graph generative models (A, B, C) over a range of values of $\sigma$ on the Erdös-Rényi Graphs dataset. Additionally, the MMD distance between the test graphs and training graphs is provided to give a meaningful sense of scale to the metric. It provides an idea of what value of MMD signifies an indistinguishable difference between the two distributions.

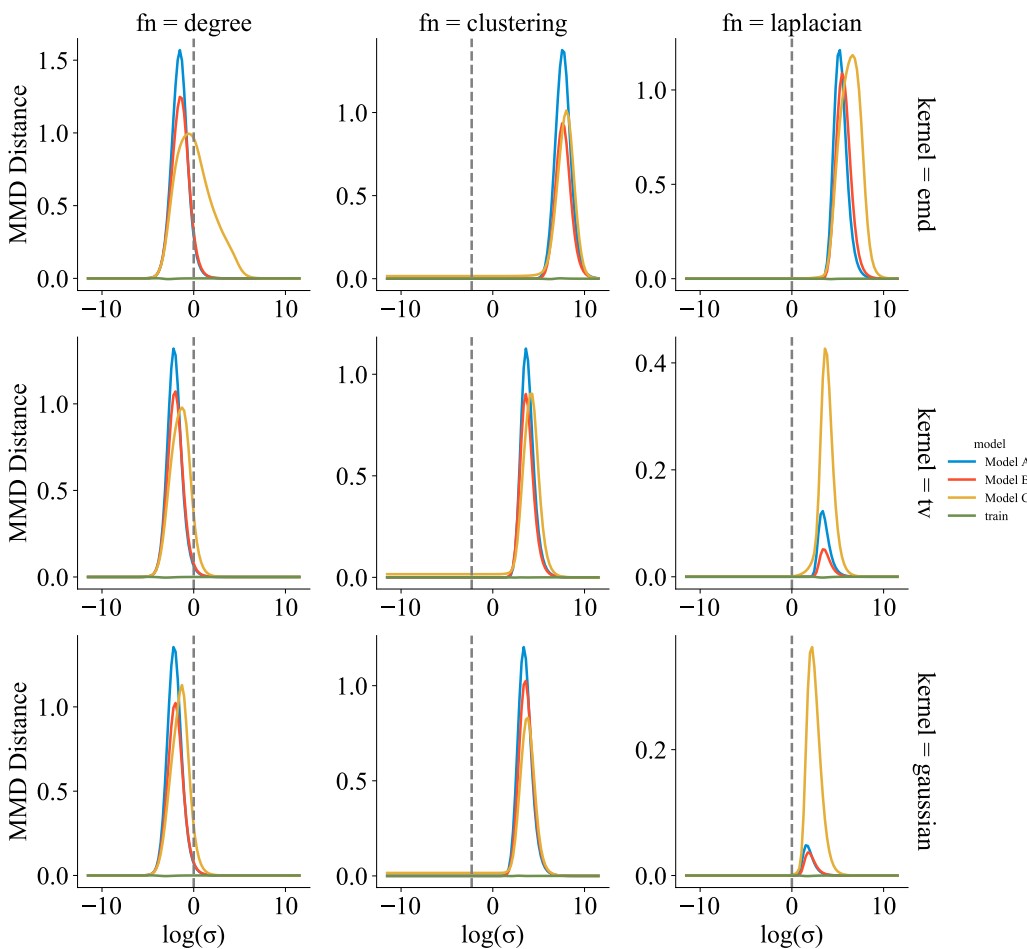

Figure 9: **Watts-Strogatz Graphs**. MMD calculated between the test graphs and predictions from three recent graph generative models (A, B, C) over a range of values of $\sigma$ on the Watts-Strogatz Graphs dataset. Additionally, the MMD distance between the test graphs and training graphs is provided to give a meaningful sense of scale to the metric. It provides an idea of what value of MMD signifies an indistinguishable difference between the two distributions.

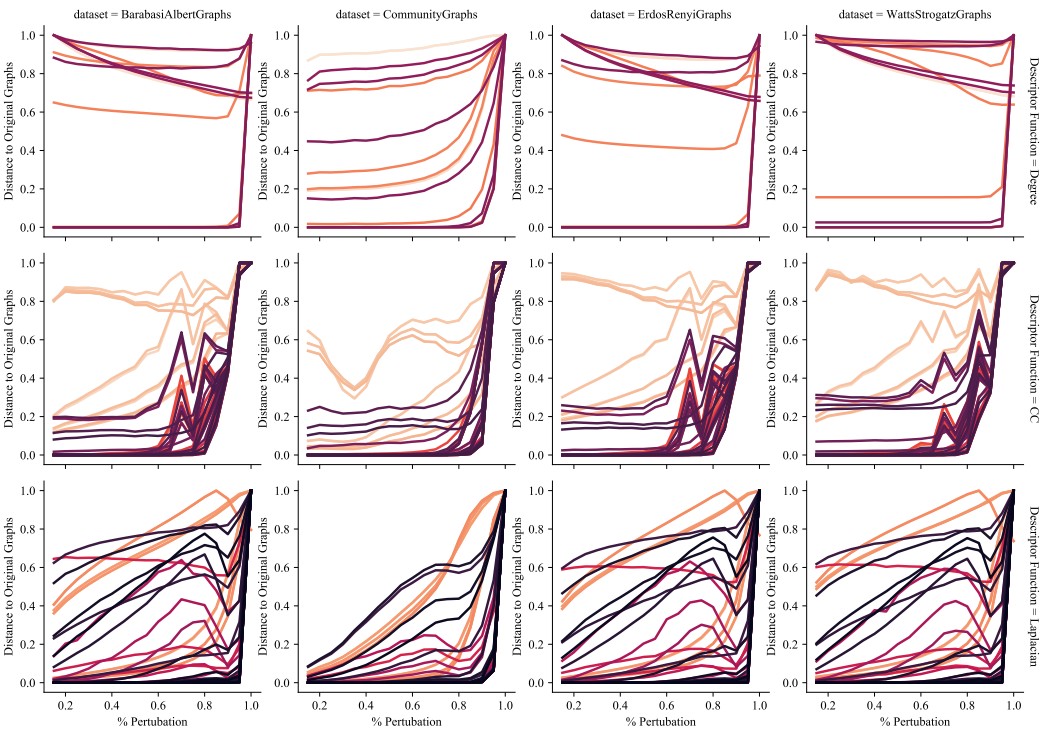

Figure 10: **Perturbation: adding edges**. This figure shows the full results across datasets, descriptor functions and parameters when the perturbation is adding edges to the graphs. At each level of perturbation, the distance of the perturbed graphs is calculated to the original graph distribution using the specified evaluator function. Each line represents a different parameter combination. An ideal evaluator function would monotonically increase as the degree of perturbation increases.

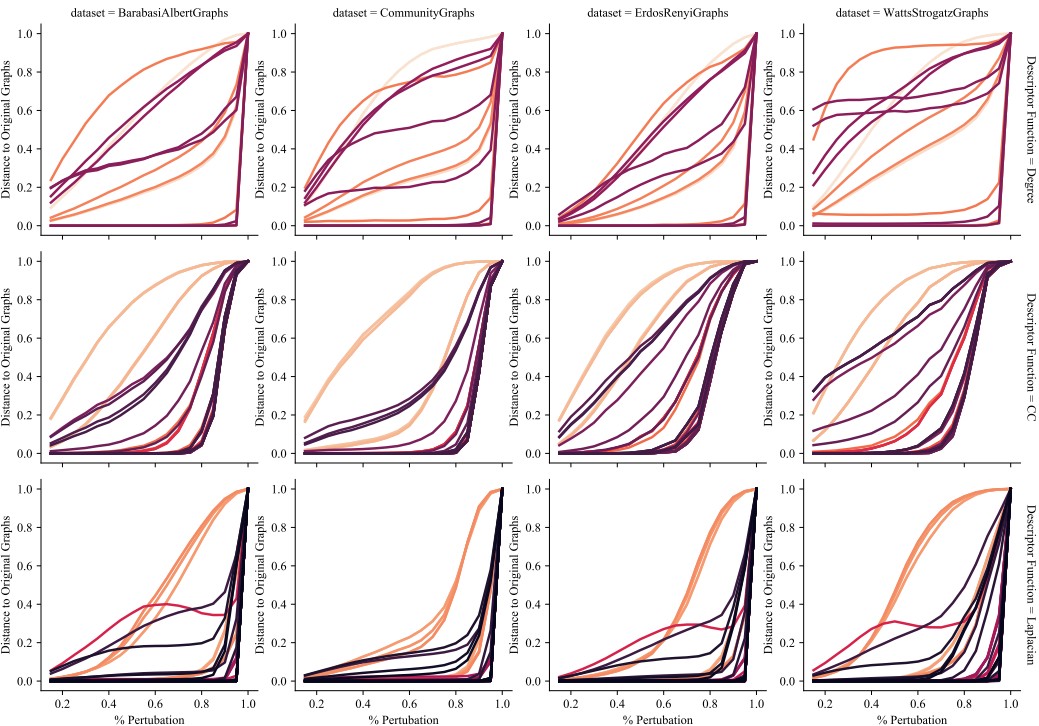

Figure 11: **Perturbation: removing edges**. This figure shows the full results across datasets, descriptor functions and parameters when the perturbation is removing edges from the graphs. At each level of perturbation, the distance of the perturbed graphs is calculated to the original graph distribution using the specified evaluator function. Each line represents a different parameter combination. An ideal evaluator function would monotonically increase as the degree of perturbation increases.

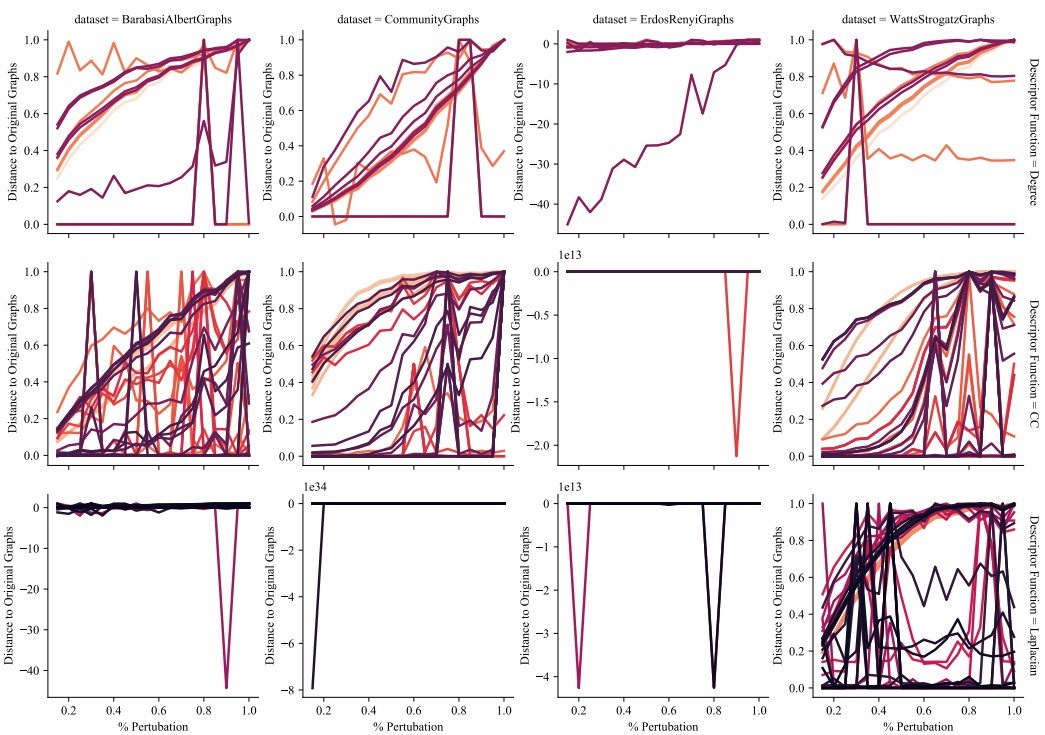

Figure 12: **Perturbation: rewiring edges**. This figure shows the full results across datasets, descriptor functions and parameters when the perturbation is rewiring edges in the graphs. At each level of perturbation, the distance of the perturbed graphs is calculated to the original graph distribution using the specified evaluator function. Each line represents a different parameter combination. An ideal evaluator function would monotonically increase as the degree of perturbation increases.

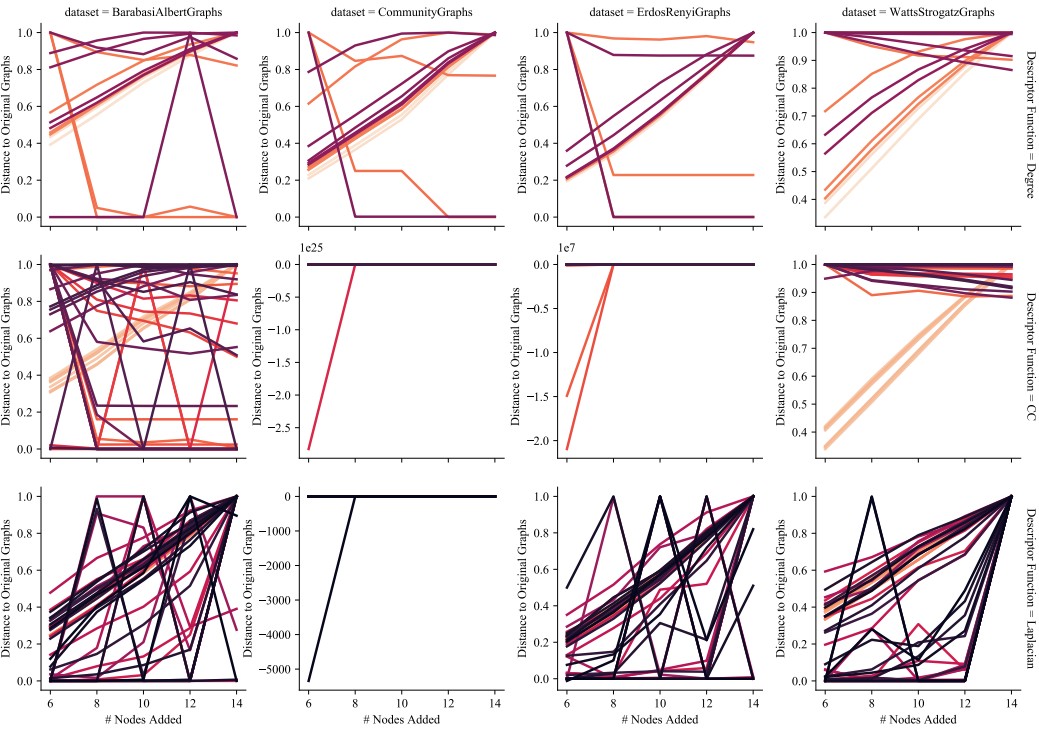

Figure 13: **Perturbation: adding connected nodes**. This figure shows the full results across datasets, descriptor functions and parameters when the perturbation is adding connected nodes to the graphs (for each node that is added, there is a 15% chance the node will be connected to any other node in the graph). At each level of perturbation, the distance of the perturbed graphs is calculated to the original graph distribution using the specified evaluator function. Each line represents a different parameter combination. An ideal evaluator function would monotonically increase as the degree of perturbation increases.

## A.7 ALTERNATIVE MEASURES OF CORRELATION

As a general recommendation, we chose to use the Pearson correlation coefficient to select a good kernel-hyperparameter combination, due to its simplicity and ease of interpretation. It reflects the behavior we expect from perturbations: as the graphs are increasingly perturbed, the distance to the original graphs should grow in a similar manner as well. However, there could be scenarios in which the distance to the original graphs should not grow linearly with the degree of perturbation, in which case the Pearson correlation coefficient would not be the best choice. We present here two plug-in alternatives for the Pearson correlation coefficient, namely the Spearman rank correlation coefficient, and mutual information, which can easily be integrated into our framework. We add one word of caution when using the mutual information, which is the fact that it does not capture the directionality of dependence. This is only an issue if there are scenarios in which the MMD distance *decreases* as the degree of perturbation increases. Since we observed this in some of our datasets, it would not be the most appropriate to use in this specific case. We now present the results from using these measures of dependence below.

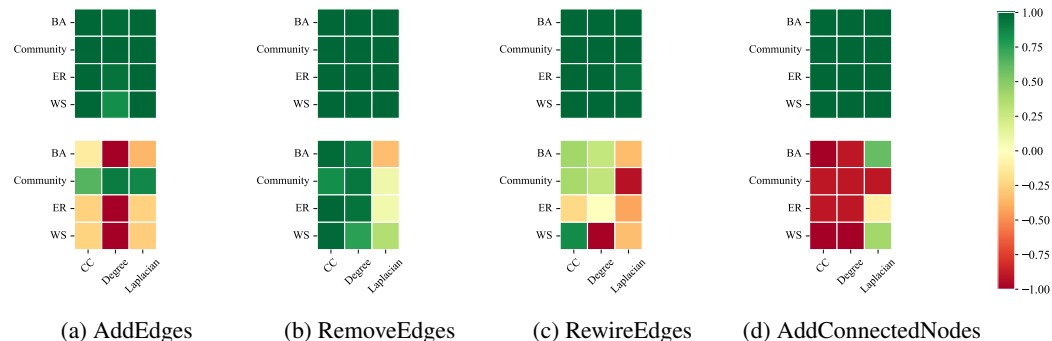

| (a) AddEdges | (b) RemoveEdges | (c) RewireEdges | (d) AddConnectedNodes |

Figure 14: The Spearman rank correlation of MMD with the degree of perturbation in the graph, assessed for different descriptor functions and datasets (BA: Barabási-Albert Graphs, ER: Erdös-Rényi Graphs, WS: Watts-Strogatz Graphs). For an ideal metric, the distance would increase as the degree of perturbation increases; resulting in a correlation close to 1. The upper row shows the *best* kernel-parameter combination in terms of the correlation; the bottom row shows the *worst*. As we can see, a proper kernel and parameter selection leads to strong correlation to the perturbation, but a bad choice can lead to inverse correlation, highlighting the importance of a good kernel/parameter combination.

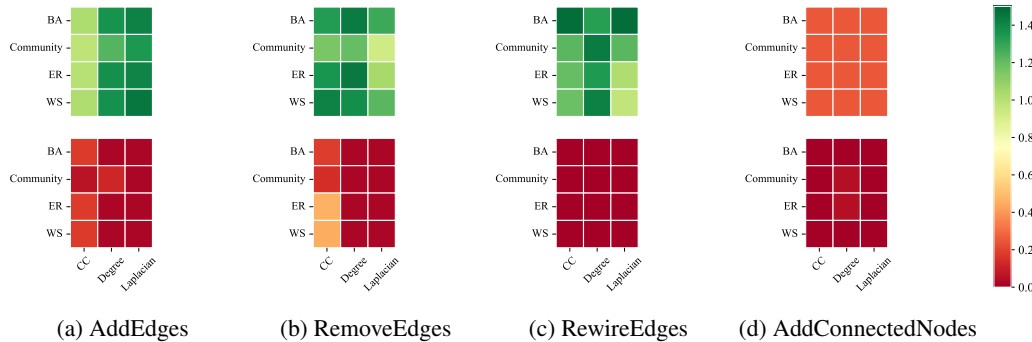

| (a) AddEdges | (b) RemoveEdges | (c) RewireEdges | (d) AddConnectedNodes |

Figure 15: The mutual information of MMD with the degree of perturbation in the graph, assessed for different descriptor functions and datasets (BA: Barabási-Albert Graphs, ER: Erdös-Rényi Graphs, WS: Watts-Strogatz Graphs). For an ideal metric, the distance would increase as the degree of perturbation increases; resulting in a high mutual information coefficient. The upper row shows the *best* kernel-parameter combination in terms of the mutual information; the bottom row shows the *worst*. As we can see, a proper kernel and parameter selection leads to strong dependence on the perturbation, but a bad choice can lead to no mutual information, highlighting the importance of a good kernel/parameter combination.

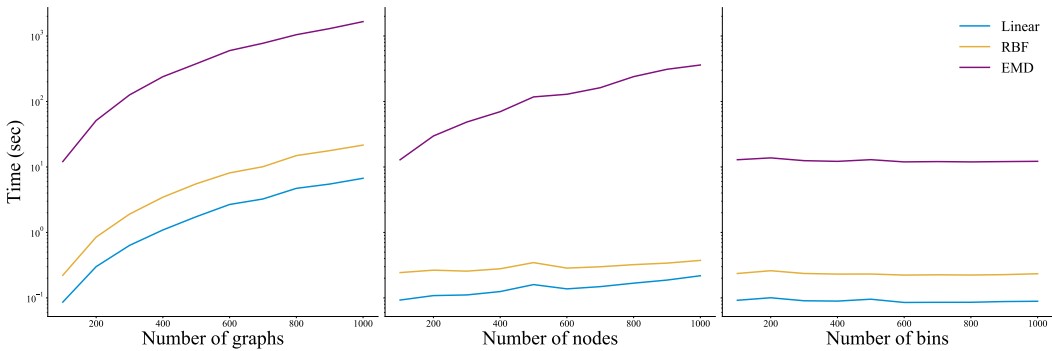

Figure 16: CPU runtime comparison (lower is better) of the linear, RBF, and EMD-based kernels when evaluated on ER graphs and varying the number of graphs (the dataset size), the number of nodes (the size of each graph), and histogram bin size. Each plot varies a single parameter (on the $x$-axis), while keeping the other two fixed with values of 100. Runtimes are reported on a logarithmic scale.

## A.8 COMPUTATIONAL RUNTIME OF DIFFERENT KERNELS

In the following, we assess the empirical CPU runtime of the linear kernel, RBF kernel, and the EMD-based kernel in the MMD calculation. As noted by ourselves and Liao et al. (2019), the EMD-based kernel is computationally expensive, hindering its suitability for use in graph generative model evaluation. We investigate this effect in more detail by doing a runtime comparison of the three kernels in a simulated environment. We generate two sets of Erdös-Rényi Graphs with the probability of an edge $p = 0.3$, and then calculate the MMD distance between the two sets using the degree distribution as the descriptor function for a varying dataset size ($n_{\text{graphs}}$), graph size ($n_{\text{nodes}}$), and histogram bin size ($n_{\text{bins}}$). As a default, we set $n_{\text{graphs}} = 100$, $n_{\text{nodes}} = 100$, and $n_{\text{bins}} = 100$. We then change one variable at a time, iterating through values $\{100, 200, \ldots, 1000\}$ while keeping the other two variables fixed, and measured the time it took to calculate the MMD distance. We report the average runtime over ten repetitions to obtain more stable results.

Our results can be seen in Figure 16. At the default setting of 100 graphs, 100 nodes per graph, and 100 bins in the histogram, we observed that the EMD-based kernel took more than 50-140 times longer to compute compared to the RBF and linear kernels respectively. This difference was exacerbated when the size of the dataset ($n_{\text{graphs}}$) increased as well as when the size of the graphs in the dataset ($n_{\text{nodes}}$) increased. For a dataset of 1,000 graphs, the EMD-based kernel took 27 minutes to run, whereas the linear kernel took 6 seconds and the RBF kernel took 21 seconds. Doing a simple extrapolation of the runtime of a dataset comprising 10,000 graphs, we estimate the EMD-based kernel would take 50 hours to run for a single hyperparameter combination, showcasing the computational limitation of this choice of kernel for larger dataset sizes. We observed a similar, albeit less extreme, increase in runtime when the size of the graphs increased. While the linear time approximation of MMD (Gretton et al., 2012a) could mitigate some of the runtime challenges, in general we recommend that practitioners use efficient kernels such as the linear kernel or RBF kernel for graph generative model evaluation.

