# OpenReview forum: "Evaluation Metrics for Graph Generative Models: Problems, Pitfalls, and Practical Solutions"
_ICLR.cc/2022/Conference — ICLR 2022 Spotlight_

### Official Review · Reviewer_7gMp · 2021-10-24

**Correctness:** 3
**Technical Novelty And Significance:** 3
**Empirical Novelty And Significance:** 3
**Recommendation:** 8
**Confidence:** 4

**Details Of Ethics Concerns:**

None identified

**Main Review:**

Strengths:
- this is a much-awaited paper that clarifies many aspects related to MMD for evaluating graph generative models. It was really interesting to see how much results may vary whenever the parameters of MMD are not calibrated properly.
- the recommendations on how to tune the MMD parameters to obtain meaningful results are really useful, and if properly incorporated into the graph generative modeling standard practices might increase the quality of the research and help establish more powerful benchmarks to evaluate graph generative models.

Weaknesses:
- while the content is extremely useful, I find the scope of the paper a bit limited. For example, node and edge features are not taken into consideration in this paper. In its current form, the paper only discusses how to use MMD to compare the structure of graphs, while in practical scenarios (e.g. for molecular generation), nodes and edge features play an equally important role.

Questions:
- commonly, MMD is also computed using the 4-orbit count descriptor. Why wasn't this taken into account in your analysis?
- I think a runtime comparison of the various kernels (at least between linear, EMD and RBF) would be something useful to the practitioner. Otherwise, claims such as "the EMD kernel is slow" remain a bit vacuous (how much slow is "slow"? At which point using EMD becomes to be intractable?)

**Summary Of The Paper:**

The paper identifies limitations and common pitfalls that practitioners encounter when using the MMD metric to evaluate graph generative models. Moreover, it provides guidance on how to ensure that the metric is properly calibrated to avoid nonsensical results.

**Summary Of The Review:**

I am inclined towards accepting the paper for its much-needed practical guidance on the use of MMD in the context of graph generation, which could have very positive implications in this research field. However, I acknowledge (and the authors should, too) that what is discussed in this paper is only one of the many aspects to consider when evaluating graph generators.

----EDIT----
I find the response of the authors to my inquiries satisfactory. For this reason, I am raising my score from 6 to 8 and vote for its acceptance.

---

> ### Author Response · Authors · 2021-11-12
> **Thank you for your review!**
>
> Thank you for your thoughtful review and helpful comments. We’ve addressed the points in the revised PDF and below. If you have any other doubts or questions we’d be happy to discuss them.
>
> > while the content is extremely useful, I find the scope of the paper a bit limited. For example, node and edge features are not taken into consideration in this paper. In its current form, the paper only discusses how to use MMD to compare the structure of graphs, while in practical scenarios (e.g. for molecular generation), nodes and edge features play an equally important role.
>
> Thank you for your comment. We tried to address in the general section why we restricted our scope to only investigating MMD on structural aspects of the graph, since we see this as a necessary step to be solved first, since evaluating node/edge attributes also relies on having compared the structural similarity as well. We didn’t consider the use case of molecular generation since these graphs are assessed by a different set of metrics, which are domain specific and thus we felt they should be considered separately. If you have any doubts please feel free to let us know and we’d be happy to continue discussing this.
>
> > commonly, MMD is also computed using the 4-orbit count descriptor. Why wasn't this taken into account in your analysis?
>
> Calculating the orbit descriptor function is computationally very expensive (due to the counting of graphlets), and therefore didn’t satisfy the “efficiency” criteria of our desiderata. However, we plan on making our evaluation framework open source such that the community can add additional descriptor functions that are useful for their application, and thus it can easily be integrated into our framework if desired. We’ve added a statement to Section 3.1 to clearly state that this descriptor is sometimes used, but is not considered in depth in the paper due to the computational cost of it.
> > I think a runtime comparison of the various kernels (at least between linear, EMD and RBF) would be something useful to the practitioner. Otherwise, claims such as "the EMD kernel is slow" remain a bit vacuous (how much slow is "slow"? At which point using EMD becomes to be intractable?)
>
> This is a great idea! We’re currently running this experiment, and will update  the PDF once we have the results. In the mean time, we just mention that Liao et al. (2019) also noted how slow EMD was to compute on moderately large graphs, which is what drove them to use the (non-psd.) kernel using the total variation distance. We’ll update this thread once we’ve revised the PDF with these results.
>
> > I acknowledge (and the authors should, too) that what discussed in this paper is only one of the many aspects to consider when evaluating graph generators.
>
> This is a very good point, thank you for highlighting it. We clarified that we are only investigating the evaluation of the structural aspects of the sets of graphs in the Introduction, and that future work will need to investigate assessing both aspects of graphs.

---

> > ### Comment · Reviewer_7gMp · 2021-11-12
> > **Thanks for the response**
> >
> > I read the revised version. Personally, I am quite satisfied with the final results, I think you did a good job at accomodating my requests and those of the other reviewers. I look forward to the runtime comparison experiment. In the meantime, I decided to increase my score to an 8. Fingers crossed to see you published at ICLR!

---

> > > ### Author Response · Authors · 2021-11-18
> > > **Thanks for the reply; runtime experiments have been added**
> > >
> > > Thank you for your reply and your support! We finished the runtime comparisons and have included them in the revised PDF. We added a reference to the runtime results in the paragraph *Choose valid and efficient kernel candidates*, and have provided the full results in Appendix A.8.

---

### Official Review · Reviewer_d5fz · 2021-11-02

**Correctness:** 3
**Technical Novelty And Significance:** 3
**Empirical Novelty And Significance:** 3
**Recommendation:** 8
**Confidence:** 4

**Main Review:**

Strengths :-
1) The problem the authors focus on in this work is significant in graph based generative modeling community. By highlighting the potential issues with usage of the MMD metric, the authors provide valuable insight and learning which would hopefully advance the state of research.
2) The notation and description of the paper is unambiguous and clear and the authors via rigorous empirical study highlight different facets of the problem effectively. By not deciding to include their own approach, the authors add to the clarity of the paper for which I would like to commend their work. Additionally most of the experiments are reproducible which is another reason to recommend this work.
3) The recommendations from the authors is insightful and would be useful to researchers working in the field of graph based generative models.

Weaknesses :-
1) The current scope and focus of the paper is a bit limited. There are graph based generative models which focus on edge prediction and node classification tasks for example which focus on graph based convolutions using node and edge based features and also use common metrics like logistic loss etc. There are also other forms of graph based generative models which try to learn node and edge based embeddings via deep neural networks etc.
2) The authors could have tried to include other metrics in common usage apart from the MMD metric which would have made this work even more beneficial. Same goes for other prevalent descriptor functions.
3) Rather than working with correlation which focusses on linear relationships, the authors could have worked with non-linear variants such as mutual information which would potentially have been more apt.

**Summary Of The Paper:**

In this work, the authors aim to provide a principled way to evaluate and compare graph generative models. The authors initially list desirable criteria an evaluation metric should possess and subsequently discuss the usage of maximum mean discrepancy (MMD) for model comparison. Subsequently they highlight issues with the usage of the MMD metric for model comparison purposes and via empirical study and analysis, list a set of practical recommendations for researchers to follow when evaluating graph based generative models.

**Summary Of The Review:**

The authors focus on an important problem in the graph based generative modeling domain and provide valuable insight and learnings via carefully designed empirical results and clear and unambiguous explanations and intuitions. The recommendations from the authors is insightful and would be useful to researchers working in the field of graph based generative models. However the current scope and focus of the paper is a bit limited. The authors could have expanded this work via considering i) other metrics apart from MMD, ii) other descriptor functions as well as iii) working with non-linear variants for correlation, i.e., mutual information. I would like to congratulate the authors for this work and ideally would like to see this paper accepted.

----UPDATE----

After revisiting the updates made by the authors, I am updating my score to Accept. I would like to congratulate the authors for their excellent work which would benefit researchers working in the field of graph based generative models.

---

> ### Author Response · Authors · 2021-11-11
> **Thank you for your review!**
>
> Thank you very much for your encouraging words and helpful feedback, which we have tried to address below and in the revision. If you have any remaining questions or doubts, please don't hesitate to let us know.
>
> > The current scope and focus of the paper is a bit limited. There are graph based generative models which focus on edge prediction and node classification tasks for example which focus on graph based convolutions using node and edge based features and also use common metrics like logistic loss etc. There are also other forms of graph based generative models which try to learn node and edge based embeddings via deep neural networks etc.
>
> This is absolutely right, and we hope to address this in future work. We restricted our investigation to models that are predicting structural properties of graphs, since more sophisticated models that predict node/edge labels also need a way to assess the structural aspect of the generative model, and we found that widespread practice had some notable flaws in just the most basic, fundamental aspect. We hope to help the field improve in that aspect, so that we can later go on to tackle the more sophisticated scenarios going forth.
>
> > The authors could have tried to include other metrics in common usage apart from the MMD metric which would have made this work even more beneficial. Same goes for other prevalent descriptor functions.
>
> We fully agree that the next step is to develop additional metrics or descriptor functions that would be meaningful to compare graph generative models. We are actively working on the development of new methods for comparing graph generative models, and we consider this paper to be a crucial first step or foundation on top of which additional works can be published.
>
> > Rather than working with correlation which focuses on linear relationships, the authors could have worked with non-linear variants such as mutual information which would potentially have been more apt.
>
> We didn’t use mutual information because it doesn’t capture the directionality of the dependence, and we observed several cases where increasing perturbations led to a decrease in the MMD distance (which is not ideal behavior). In this scenario, MI reported high dependence because there was a relationship between the degree of perturbation and the distance. We ultimately chose linear correlation because of its interpretability and the clear expected relationship between the degree of perturbation and the degree of dissimilarity. It is, however, possible that in certain situations the degree of perturbation could result in non-linear increases in dissimilarity, and other correlation measures may be more appropriate. We’ve augmented the correlation analysis to include the Spearman’s rank correlation and mutual information (for cases where no decrease in distance is observed) in addition to Pearson’s correlation. While we still show the Pearson's correlation coefficient in the main text for the reasons mentioned above, but we now refer the reader to a new section A.7 in the Appendix which discusses these alternative options in more detail.

---

### Official Review · Reviewer_37EN · 2021-11-02

**Correctness:** 3
**Technical Novelty And Significance:** 3
**Empirical Novelty And Significance:** 3
**Recommendation:** 8
**Confidence:** 5

**Main Review:**

The paper has several strengths.

1) Presentation. The paper is well written even for people that are not related to this topic. The paper focuses on the main pitfalls of this measure.

2) Reproducibility: Most of the experiments can be easily replicated. However, there is some lack of explanation in the main paper (9 pages) that could be addressed.

3) Impact of ideas: The criticism of this measure could have a great impact on the current evaluation process, changing the way that GGMs must be evaluated.

4) Technical quality: Most pitfalls are corroborated empirically, while others are not even necessary to check.

Even though the main strengths of the paper, there are some issues that reduce the recommendation score. unfortunately, the solutions are vaguely explained, and not necessarily new.

1) Perturbations are not defined in the paper (9 pages). This makes it difficult to understand some of the experiments and the effects in the final evaluation. Please include part of the appendix in the main part of the paper.

2) The difference in the distributions versus perturbations is not necessarily a problem of the metric, it could be the descriptor function. The paper uses the clustering coefficient to show this problem; however, depending on the graph, the clustering coefficient does not change considerably with some type of perturbations. For example, imagine a graph with a low clustering coefficient (as many real-world networks). If we pick a node with cc=0 and we remove all of its edges, its cc will not change at all; even though it was highly perturbed. Please, change the example, using a measure affected by this issue or define that this problem is given to some descriptor functions.

3) Provide a sense of scale: This idea is vaguely explained, and it does not define a procedure to follow. Please, define a procedure.

4) Utilize meaningful descriptor functions: This does not show a new solution. There are several meaningful descriptor functions that are used in the evaluation of GGM that are not even included in this paper. For example, the geodesic distance. Expand the use over other meaningful descriptor functions.

5) One of the main problems, interpretability, is not even solved.

6) There is no comparison against other measures. For example, the multivariate Kolmogorov-Smirnov, that has been used in a previous evaluation of GGMs (A multivariate Kolmogorov-Smirnov test of goodness of fit and Tied Kronecker Product Graph Models to Capture Variance in Network Populations)

**Summary Of The Paper:**

The paper criticizes Maximum Mean Discrepancy (MMD) an evaluation metric that has been used lately to evaluate Generative Graph Models (GGMs). The main problems of the MMD are: It does not capture difference upon perturbations; It does not have a scale; It requires the selection of a kernel and parameters that could lead to different results (and a high time complexity according to the kernel). Finally, the paper describes some procedures based on MMD to avoid some of these problems. While the pitfalls are clearly explained, the solutions are vague.

**Summary Of The Review:**

The paper mainly focuses on the negative problems of the MMD, which are explained and in several cases empirically proved. This makes the paper strong and it should be accepted. However, the solutions are not clearly explained, decreasing the score for this paper.

---

> ### Author Response · Authors · 2021-11-11
> **Thank you for your review!**
>
> Thank you for your helpful feedback; we’ve addressed them in the revised paper. If you have any remaining questions, please let us know.
>
> > Perturbations are not defined in the paper...Please include part of the appendix in the main part of the paper.
>
> We’ve added a new section in 5.1 called *Creating dissimilarity via perturbations*, where we explain how we perform the perturbations. We describe the process and refer to the formal definitions in the Appendix (due to space constraints). Please let us know if you think it still needs more detail.
>
> > The difference in the distributions versus perturbations is not necessarily a problem of the metric, it could be the descriptor function. The paper uses the clustering coefficient to show this problem; however, depending on the graph, the clustering coefficient does not change considerably with some type of perturbations. Please, change the example
>
> We fully agree; in some situations, like the one you described, certain descriptor functions are not meaningful and thus should not be used. We mentioned the following example in the paragraph *MMD does not always capture differences in distributions* to highlight that a descriptor function (here, specifically the CC) may be the problem, and not MMD, and thus should not be used:
>
> *While one can construct specific scenarios in which two distributions become farther apart but the distance does not monotonically increase (for instance, removing edges from a triangle-free graph, and using the clustering coefficient as the descriptor function), in such cases, the specific choice of descriptor function $f$ is crucial to ensure that $f$ can capture differences in distribution.*
>
> Figure 2b only shows the result of the clustering coefficient due to space constraints, but the same plot is shown for all descriptor functions &  datasets in Figure 10 of Appendix A.6. The clustering coefficient on the  Community Graphs dataset captured all the potential problems in one plot (sometimes the MMD distance stays flat until nearly fully perturbed, sometimes the MMD distance decreases, etc), which is why we selected that one, to maximize the information given the limited space we had.
> We wanted to explain our logic and then check if you’d still prefer a different example? We can easily change it if you think a different example would be better.
>
> > Provide a sense of scale: This idea is vaguely explained, and it does not define a procedure to follow. Please, define a procedure.
>
> We updated the section *Provide a sense of scale* in Section 5. Please let us know if this did not help to clarify the procedure to follow, and we can think more on how to rewrite this to make it clearer.
>
> > There are several meaningful descriptor functions that are used in the evaluation of GGM that are not even included in this paper. For example, the geodesic distance. Expand the use over other meaningful descriptor functions.
>
> The paper was focused (due to page limit constraints) on investigating the tools that are in use today. Among the graph descriptor functions being used (with the exception of the orbits, which were computationally too expensive), we didn’t find anything wrong with the functions in use today, and thus found they were all sufficient to use (unless a domain specific choice mandates otherwise). We investigated the descriptor functions we found most frequently in use (You et al. (2018); Liao et al. (2019); Niu et al. (2020); Goyal et al. (2020); Dai et al. (2020); Zhang et al.(2021); Chen et al. (2021); Mi et al. (2021); Podda & Bacciu (2021)), which is why we didn’t include the geodesic distance, or others, since we did not find others that were  commonly used in multiple papers. We fully support further research into novel descriptor functions (but due to practical reasons could not investigate this to the extent needed within the page limit), and have added the geodesic distance into the outlook as an example descriptor function to investigate. If you wouldn’t mind sharing the paper that uses the geodesic distance, we’re interested to look more into it!
>
> > One of the main problems, interpretability, is not even solved.
>
> The evaluation process does not give much insight into what the graph generative models do, and interpretability of graph generative models is still an open question. Did you mean interpretability of the evaluation? Is there anywhere in the text we could better clarify this?
>
> > There is no comparison against other measures. For example, the multivariate Kolmogorov-Smirnov, that has been used in a previous evaluation of GGMs
>
> We did not find the multivariate KS to be widely adopted among recent graph generative model papers, and for this reason we did not investigate it in detail in our paper, but we think this could be a potential alternative to using MMD, and fully encourage its investigation in future work. We’ve added a reference to this in our outlook; thank you for the suggestion!

---

### Official Review · Reviewer_3VAJ · 2021-11-06

**Correctness:** 3
**Technical Novelty And Significance:** 3
**Empirical Novelty And Significance:** 2
**Recommendation:** 5
**Confidence:** 3

**Main Review:**

This paper is generally clear and well written.

It could be helpful to add citations in a few places to help the reader who is not already familiar enough with the graph generative models community to recognize what is considered most popular and status quo (see minor comments below for some examples).

One limitation to the robustness experiments is that there is an assumption that a greatly perturbed graph should have a much different probability than the source graph, however it is not clear why this is necessarily the case (e.g, if the graph generating distribution is uniform).

The paper also notes and makes a point in the experiments how three popular works all use different kernels, descriptors, hyperparameters, however one of the works (Niu et al., 2020) doesn’t seem to describe the kernel and hyperparameters, instead citing one of the other of the other two works (You et al., 2018) and claiming to use the same evaluation (so presumably the same kernel and hyperparameters).  In addition, all three works appear to use degree histogram, clustering coefficient histogram, as well as the number of orbits with 4 nodes (which was left out from this paper); (Liao et al, 2019) appears to uses the Lapacian spectra in addition to these.  It might be worth noting why the number of orbits was omitted in this study.

Minor comments:
- “the community has largely gravitated towards a single comparison metric, the maximum mean discrepancy (MMD)” consider adding some citations (and if there are notable exceptions)
- “Specifically, d(G, G′) should be monotonically increasing the further apart G and G′ are from one another.” Further apart according to what measure? Presumably there are several ways to specify this and, if taken precisely, might beg the question of how to define d.
- “if a distribution G is subject to a perturbation, a suitable metric should be robust to small perturbations” is it correct to understand this as the metric being well-conditioned?
- rather than perturbing graphs (since the resultant graphs might still be in the set of graphs or have high probability), why not generate sets of graphs from increasing different distributions directly?
- “depicts an overview of this approach as it is being done today.” Consider adding citation(s).
-  "Equation 1 is often treated as a metric as well” consider adding citations
- “even though it would be more correct to only use it for hypothesis/two-sample testing” the original (and several subsequent) works seem to describe MMD in the hypothesis/two-sample testing context, but it might still be worth adding a brief note why treating it as a metric is "less correct".
- It seems as though there’s a typo in (2)?  The numerator appears to be 2 * deg(v), when I believe it should be a count of the edges between neighbors of v (instead of v's degree)?
- “and only require further analysis of their expressivity and robustness” given this work's nature and scope, this would be nice to address in this work
-  The paper notes how descriptor functions are strictly necessary for MMD, and when they are used, the importance of descriptor function choice.  In the recommendations section, the suggestion appears to be to use any of the descriptors (as opposed to how to select an/the appropriate one).
-“clustering coefficient (CC), is even negatively correlated with the degree“, the scale in Fig 4 (correlation) appears to be 0 to 1 instead of -1 to 1?

**Summary Of The Paper:**

This paper describes desiderata (expressivity, robustness, and efficiency) for metrics for comparing graph generative models and details the various ways that recent work has used maximum mean discrepancy (MMD) to evaluate graph generative models.  Several limitations of MMD are described, as well as the consequences of the various choices in using MMD (kernels, descriptor functions, and hyper-parameters) and how these choices have impacted experiments in recent work.  The paper concludes with recommendations to help ameliorate several of these limitations and impacts.

**Summary Of The Review:**

The paper is generally clear and well written and makes several interesting and worthwhile observations about MMD and its use in recent work.  However, there is an opportunity to better support some of the claims--like what is considered status quo (through citations) and about recent experiments (which paper used which settings and, in the case of (Niu et al., 2020), where those setting can be found--and to clarify the perturbation based desiderata and experiments/analysis.

---

> ### Author Response · Authors · 2021-11-12
> **Thank you for your review!**
>
> Thank you for your thoughtful review & your actionable feedback, which we gladly incorporated. If you have any other questions or doubts, please let us know.
>
> > Niu et al., (2020) doesn’t seem to describe the kernel & hyperparameters
>
> The three papers adapt the setup from You et al. (2018), including the hyperparameters (HP), which is surprising since each paper used a different kernel. The HP were not provided; we found them in their code. We provide a summary of the values that the authors used in Table 1, but think these choices should be stated clearly and chosen according to a meaningful procedure (such as the one we suggest).
>
> > [Note] why the number of orbits was omitted.
>
> We have added it to Section 3.1. Calculating the orbit descriptor function is computationally very expensive (due to the counting of graphlets), & therefore didn’t satisfy the “efficiency” criteria of our desiderata. However, it could easily be integrated into our open source framework if desired.
>
> > “the community has largely gravitated towards a single comparison metric, the maximum mean discrepancy (MMD)”....“depicts an overview of this approach as it is being done today.”... "Equation 1 is often treated as a metric as well”...Consider adding citation(s).
> Thank you, we’ve added 9 references.
>
> > “Specifically, d(G, G′) should be monotonically increasing the further apart G & G′ are from one another.”...according to what measure?...might beg the question of how to define d.
>
> This is the million dollar question! Ideally we would use the graph edit distance, but this is computationally infeasible. Thus the community is defining some “proxy” for this, e.g. by using graph descriptor functions combined with MMD. We hope that highlighting the shortcomings of the status quo will spark new research into better ways to define d.
>
> > a...metric should be robust to small perturbations” is it correct to understand this as the metric being well-conditioned?
>
> Ideally, we would consider a metric to be stable if, for a small perturbation $\epsilon$ of two graphs $G$ & $G'$ (for instance, the addition of an edge), the metric satisfies $d(G, G') < \delta(\epsilon)$, i.e. it is upper-bounded by a constant $\delta$, which may depend on the strength of the perturbation.
>
> > rather than perturbing...why not generate sets of graphs from increasing different distributions directly?
>
> This is feasible to do with synthetic graphs, such as ER, where the difference in distribution is controlled by a parameter (e.g. the probability of an edge). Different datasets have different generation properties & parameters, so there is no single way to generate graphs from different distributions across different datasets. It gets particularly tricky when using real world datasets, because there is no ground-truth or single parameter that clearly delineates what is in or out of distribution. Since we want to generate graphs that are increasingly different in a consistent way across all kinds of graphs, systematic perturbations were the only consistent, controlled way we could confidently deem two distributions as different from one another.
>
> > “it would be more correct to only use it for hypothesis/two-sample testing”...[add] a brief note why treating it as a metric is "less correct".
>
> Gretton et al. (2012) themselves noted in their MMD paper that the empirical estimate of MMD can take negative values, since it is an unbiased estimator. (We observed this phenomenon in our own experiments as well). There are certain conditions under which MMD forms a metric, but if any of the conditions are not satisfied, it may not be a proper metric, e.g. the triangle inequality might not hold, which could result in spurious ranking comparisons. We added some explanation to the text to provide justification of the statement to make it clearer, thank you!
>
> > It seems as though there’s a typo in [Eq.] (2)?
>
> Thank you for pointing this out, we’ve corrected the notation!
>
> > “and only require further analysis of their expressivity & robustness”...this would be nice to address in this work
>
> In general, robustness of kernel-based methods is still an active research topic (https://arxiv.org/abs/1605.09522) & a bit beyond what we can achieve in this work.  We’ve clarified that the RBF kernel is universal (at least on compact sets of R^d), making it a suitable choice for MMD (because it guarantees injectivity of the mean embedding representation).
>
> > the suggestion appears to be to use any of the descriptor [functions] (as opposed to how to select an/the appropriate one).
>
> The paper focused (due to low space) on investigating the functions in use today. Besides orbits, we didn’t find anything specifically wrong with them, but we fully support further research into novel descriptor functions.
>
> > the scale in Fig 4 [should be] -1 to 1?
>
> Thank you for spotting this! We generated the colorbar separately from the heatmaps, which reset the values to the default (0 to 1). We’ve corrected this.

---

### Author Response · Authors · 2021-11-11
**Thank you for your feedback!**

We’re happy that you find our work clear and well written (R1, R2, R3), and that you would like to see our work accepted (R2, R3, R4) for the insights about the behavior of MMD (R1, R2, R3, R4) and the useful recommendations for the practitioner (R3, R4) about how to properly use MMD. We were thrilled that you believe our work will have positive implications in our research field (R2, R3, R4)!

We would like to thank the reviewers for your encouraging words and for providing such helpful and actionable feedback, which we felt has strengthened our paper. While we will address specific comments in the individual reviews (and in the revised PDF, where the changes can be seen in blue), we wanted to address one general point of feedback here about the extension of the scope of the paper. There were three aspects of this that surfaced in the reviews, namely that we didn’t consider other (i) metrics/descriptor functions, (ii) we didn’t consider the evaluation of node/edge attributes, (iii) we didn’t investigate the case of molecular generation (this is of course related to (ii)). This was by design (in large part due to page limit constraints), and we would like to provide an explanation of our reasoning.

**(i) Considering other metrics/descriptor functions**: The reason for not presenting additional metrics/descriptor functions was a purely practical one.  We have started developing new metrics/descriptor functions and had initially intended to incorporate them in this work, but we found that we were not able to give *sufficient attention* to each aspect (i.e. pitfalls of the status quo, and a thorough investigation of alternative options) when trying to do both. Rather than addressing these two components simultaneously (pitfalls of MMD and providing alternative metrics), which would result in addressing both only superficially, we decided to separate them into two separate works: this first paper is intended to provide the foundation of what is currently being done and what challenges are associated with it, providing the necessary insights on why the status quo has some challenges, and how to appropriately leverage the descriptor functions & MMD. In future work, we plan on referencing the findings of this paper to provide the motivation for presenting alternative solutions. More generally, our ultimate goal for this paper is to spark a discussion about this topic in the community and stimulate research of alternative methods of comparing graph generative models.

**(ii) Evaluating node/edge attributes**: We have not focused on evaluating node/edge attributes in our work in order to first establish a foundation for assessing the structural aspects of graph generative models, and have updated the text to make our scope clearer. Jointly assessing the structural and node/edge feature distributions is a natural next step. We are aware of research that generates node/edge features and evaluates their success using a combination of MMD on the structural aspect of the graph, followed by MMD on the node/edge attribute distributions (Goyal et al, 2020). Our framework can be easily extended to account for this. We fully agree that this is crucial for the community to be working on, and hope that our work can provide the foundation for later assessing these components jointly.

**(iii) Assessing molecule generation**: Related to (ii) is the special case of molecule generation, which is a very important subclass of graph generative models. The evaluation in this case however is a bit specialized, and has its own set of evaluation metrics specific to that use case, since there are more constraints on the problem (e.g. whether the graph constitutes a valid molecule). Given the differences in how the generated graphs are evaluated  (which also has its own challenges, as can be seen in Renz et al. (2019), for instance), we consider this to be a different problem, and this is why we deliberately choose not to discuss it in depth in this publication.

We are extremely grateful for your thoughtful comments, and we’re more happy to continue the discussion if you have any other questions or doubts!

---

### Decision · Program_Chairs · 2022-01-20

**Decision:**

Accept (Spotlight)

**Comment:**

This paper provides an overview of evaluating graph generative models (GGMs). It systematically evaluates one of the more popular metrics, maximum mean discrepancy (MMD). It highlights some challenges and pitfalls for practitioners and suggests some ways to mitigate them. The reviewers found the paper practically relevant and several reviewers upgraded their scores through the discussion process. The authors acknowledged there are still some remaining issues regarding (i) considering other metrics & descriptor functions; ii) evaluating node/edge attributes and iii) addressing molecule generation. I am satisfied that these areas are beyond the scope of the current work and that the clarification improvements in the paper are adequate. It stands well enough on its own to accept in its present form.